# HS, an Ancient Molecular Recognition and Information Storage Glycosaminoglycan, Equips HS-Proteoglycans with Diverse Matrix and Cell-Interactive Properties Operative in Tissue Development and Tissue Function in Health and Disease

**DOI:** 10.3390/ijms24021148

**Published:** 2023-01-06

**Authors:** Anthony J. Hayes, James Melrose

**Affiliations:** 1Bioimaging Research Hub, Cardiff School of Biosciences, Cardiff University, Cardiff CF10 3AX, Wales, UK; 2Graduate School of Biomedical Engineering, University of New South Wales, Sydney, NSW 2052, Australia; 3Raymond Purves Laboratory, Institute of Bone and Joint Research, Kolling Institute of Medical Research, Royal North Shore Hospital, St. Leonards, NSW 2065, Australia; 4Northern Sydney Local Health District, Royal North Shore Hospital, St. Leonards, NSW 2065, Australia

**Keywords:** heparan sulfate, heparan sulfate proteoglycan, sulfation motif, cell/matrix interaction, tissue homeostasis

## Abstract

Heparan sulfate is a ubiquitous, variably sulfated interactive glycosaminoglycan that consists of repeating disaccharides of glucuronic acid and glucosamine that are subject to a number of modifications (acetylation, de-acetylation, epimerization, sulfation). Variable heparan sulfate chain lengths and sequences within the heparan sulfate chains provide structural diversity generating interactive oligosaccharide binding motifs with a diverse range of extracellular ligands and cellular receptors providing instructional cues over cellular behaviour and tissue homeostasis through the regulation of essential physiological processes in development, health, and disease. heparan sulfate and heparan sulfate-PGs are integral components of the specialized glycocalyx surrounding cells. Heparan sulfate is the most heterogeneous glycosaminoglycan, in terms of its sequence and biosynthetic modifications making it a difficult molecule to fully characterize, multiple ligands also make an elucidation of heparan sulfate functional properties complicated. Spatio-temporal presentation of heparan sulfate sulfate groups is an important functional determinant in tissue development and in cellular control of wound healing and extracellular remodelling in pathological tissues. The regulatory properties of heparan sulfate are mediated via interactions with chemokines, chemokine receptors, growth factors and morphogens in cell proliferation, differentiation, development, tissue remodelling, wound healing, immune regulation, inflammation, and tumour development. A greater understanding of these HS interactive processes will improve therapeutic procedures and prognoses. Advances in glycosaminoglycan synthesis and sequencing, computational analytical carbohydrate algorithms and advanced software for the evaluation of molecular docking of heparan sulfate with its molecular partners are now available. These advanced analytic techniques and artificial intelligence offer predictive capability in the elucidation of heparan sulfate conformational effects on heparan sulfate-ligand interactions significantly aiding heparan sulfate therapeutics development.

## 1. Introduction

Heparan sulfate (HS) is an ancient molecule of the glycocalyx that evolved early in metazoan evolution [1,2]. Within a 500-million-year period, it acquired immense molecular diversity [3], allowing it to function in molecular recognition and information storage and transfer, and developed interactive and instructive properties capable of modulating diverse cellular behaviour and physiological processes [4]. The molecular diversity in HS is a property of its glyco-code, i.e., the information encoded in the structure of its glycosaminoglycan (GAG) chains that facilitates highly specific interaction with a broad range of soluble and structural proteins. Cell-based glycan arrays and glyco-genomic profiling are now being used to define the structural determinants operative in glycan-protein interactions in living cells to better understand the diverse functional properties conveyed by HS [5]. The persistence of HS over the millennia provides strong evidence of its critical roles in many essential physiological life processes, especially when significant genetic investment in HS biosynthetic enzymes is required to produce HS. HS, at the most simplistic structural level, is composed of the repeat disaccharide GlcNAc-GlcA, however, strings of these residues also undergo modification by sulfation and acetylation/de-acetylation at variable positions along the HS GAG chain and epimerization of some of the GlcA residues to IdoA introducing a significant level of structural complexity, charge heterogeneity and interactive capability (Figure 1a–c). HS is the most heterogeneous GAG and, after heparin, it is the most highly charged [6]. Like heparin, HS contains variable regions of high sulfation (S domains), high acetylation (A domains) but also has less-modified de-acetylated regions along the HS chain [6] (Figure 1a). Acetylated regions in heparin are less extensive and occur as single residues rather than as acetylated blocks however, overall, heparin is more extensively modified than HS. Low sulfation forms of HS also occur. Analysis of the HS biosynthetic enzyme distribution in tissues correlates with the diversity of HS forms found in tissue development and maturation (Figure 2a). The sulfation of other GAGs such as dermatan sulfate (DS) and keratan sulfate (KS) are also not uniform along the GAG chain and this also equates with the interactive capability of these GAGs (Figure 2b). The development of several monoclonal antibodies (mAbs) to native chondroitin sulfate (CS), HS and keratan sulfate (KS) high and low sulfation motifs has also identified specific localizations of sulfation motifs along CS, HS, and KS chains. Specific immunolocalisation of these CS motifs in a range of tissues demonstrates they have roles in tissue morphogenesis and are associated with progenitor stem cell populations and tissue development, as well as tissue repair processes [7,8,9,10,11,12,13,14,15]. Sulfate groups are also prominent interactive features of the HS molecule, and are variably located on the HS chain (Figure 1b,c). Assembly of HS requires the coordinated activity of over 20 biosynthetic enzymes [16]. The investment by cells in such sophisticated genetic biosynthetic machinery with its accompanying metabolic energy demands shows the functional importance conveyed by HS in essential life processes. Listings in the HS interactome show that HS binds greater than 400 bioactive proteins [17]. A murine study of acute pancreatic disease, identified 786 HS-binding proteins, further expanding the HS interactome [18,19,20]. Such HS-binding proteins include a diverse range of growth factors, neurotrophins, cytokines, morphogens, chemokines, structural extracellular matrix (ECM) proteins, cell adhesion molecules, proteases, and protease inhibitors [21]. The reference manual Essentials of Glycobiology (https://www.ncbi.nlm.nih.gov/books/NBK579918/ accessed on 18 November 2022) is an extensive reference piece for glycobiology studies.

While the production of specific HS structures is an important consideration in the provision of instructive information that regulates tissue development, another consideration is that HS biosynthesis is a stochastic process. Thus, while distinct HS functional motifs have been identified, HS biosynthetic reactions have an innate ability to produce variable HS structural forms which share interactive properties through ionic interactions involving their negatively charged sulfate and carboxylate groups. Thus, while some protein–ligand interactions may involve strictly defined HS structures other interactions also occur with no apparent requirement for distinct saccharide sequences for such interactions. This raises questions about how essential the control of the production of these specific sulfation motifs are when more widespread HS interactions also occur [25]. Binding affinity and specificity are determined by the charge distribution of the sulfate and carboxylate groups and how these are presented in planar orientations [26]. Sulfate groups are bulky entities and the correct 3D spatial presentation in some cases may be an important requirement for effective receptor interactions. Epimerization of GlcA to IdoA introduces greater flexibility in the HS backbone which may allow a more extensive spatial exploration of sulfate groups in various orientations with interactive partners. Some HS interactions however may be nonspecific and ionic in nature. Thus, charge density in tissues and highly specific interactions involving rare HS structures should both be considered in the control of tissue development.

The diversity of the GAG side chains attached to PG core proteins at the cell surface contributes to their cell instructive and ECM interactive properties, allowing interaction with growth factors, chemokines, morphogens, proteases, and protease inhibitory proteins, and facilitating cell responsive cues delivered by the ECM [24,27]. The *Wnt* signalling pathway has important roles to play in embryonic tissue development and the homeostasis of adult tissue. Wnt proteins are secreted, lipid-modified glycoproteins which form morphogen gradients that direct cellular behaviour to produce these effects [28]. Wnt proteins however are hydrophobic and poorly soluble in aqueous media, heparan sulfate proteoglycans (HS-PGs) are necessary for the proper activity of Wnt proteins maintaining their solubility and these also transport Wnt proteins aiding in the formation of Wnt gradients in tissues which drive developmental processes [29]. Wnt binds to the LDLR domain II of perlecan and this acts as a Wnt delivery vehicle in tissue development [29,30]. HS-PGs are also pericellular and basement membrane ECM components and have roles in synapse stabilization in neural tissues important in synaptic plasticity and neuron interactivity within neural networks of the central and peripheral nervous systems (CNS & PNS, respectively) [24,27]. HS-PGs fine tune tissue development [31,32,33] and regulate physiological processes in mature tissues [34,35]. The diversity in the structure of the GAG side chains of HS-PGs confer cell and organ-specific functional properties and temporal and spatial variation in HS structure regulate tissue development, tumorigenesis [36] and inflammation [37]. The development of pattern-specific antibodies for HS [12,38] has enabled the demonstration of cellular and tissue distribution of HS-PGs during tissue development and of specific GAG sulfation motifs. Next-generation sequencing and state-of-the-art analytical tools now enable detailed profiling of the spatiotemporal expression patterns of GAGs and GAG biosynthetic enzymes in tissues [39,40]. In Figure 2, a heat map depicting the tissue-specific gene expression of HS biosynthetic enzymes in 37 human tissues obtained from the Human Protein Atlas [41,42] demonstrates the widespread distribution of HS-PGs in these tissues with specific HS modifications in certain tissues. Figure 2 also shows the considerable variable modifications that occur in HS in these tissues.

It is estimated that genes encoding ECM proteins represent 1.3–1.5% of mammalian genomes (http://www.pantherdb.org/genes/ accessed 3 December 2022) [43]. Transcriptome-wide identification of ECM proteins based on computational screening of >60,000 full-length mouse cDNAs for secreted proteins, followed by in vitro functional assays has recently identified several previously unidentified HS-PGs. Seven of these were basement membrane-associated PGs [43]. An investigation of the site-specific glycosylation sites of all known PGs using mass-spectrometric (MS) and glycoproteomic methodology has also identified several novel CS-and HS-PGs [44]. Of the 300–400 ECM genes estimated to be present in mammalian genomes almost one-third of these have yet to be identified [43]. Thus, the complexity of functional PGs is predicted to further increase in the future. Pikachurin and Eyes shut are two examples of HS-PGs that have been identified relatively recently and the characterization of their properties is further illustrating the sophistication of cell regulatory properties conveyed by HS-PGs in specific tissue contexts.

## 2. The Cell Instructive Properties of GAGs

As already discussed, GAGs have molecular recognition and cell instructive properties operative in tissue development, connective tissue remodelling and tissue repair [45,46]. Glycans and GAGs have key roles to play in neuronal cell function and fate [47]. KS has neuroregulatory properties [7,15,48], CS also has cell directive properties in neural crest cell migration and development of the notochord, neural tube, and neural networks [49] but generally is ascribed weight bearing properties as a component of aggrecan the major CS-PG of weight bearing tissues [50,51]. CS and the related GAG dermatan sulfate (DS) also have cell instructive properties [52,53,54,55]. These GAGs complement the biodiverse properties of HS.

### 2.1. GAG Interactions with Stem Cells

CS and HS GAGs interact with stem cells in their niche environment, promoting their escape from quiescent recycling populations and allowing them to reach pluripotency and attain a migratory progenitor cell phenotype that can participate in tissue development and repair processes [52,53,56,57,58,59]. The HS-PG perlecan is a prominent component of stem cell niches, its HS chains promote stem cell differentiation, and migratory progenitor stem cell lineages that participate in the development of a number of tissues [60,61,62,63]. In articular cartilage, the superficial tissue zone contains a stem/progenitor cell population that drives and maintains tissue growth [64,65,66,67,68]. Subpopulations of cells in this zone are associated with unique CS sulfation motif epitopes [69] and novel extracellular sulfatases, Sulf-1 and Sulf-2 [70] that may serve to regulate their differentiation status through interactions with matrix growth factors (e.g., TGF-β) thereby maintaining appositional tissue growth [71].

A highly sulfated HS glycoform is essential for the progression of mesenchymal embryonic stem cells (mESCs) to a specific terminally differentiated state that directs their development into a specified cellular lineage [72,73]. During the attainment of pluripotency mESCs initially synthesise a low-sulfated HS glycoform that drives the differentiative process [59,74] and a highly sulfated HS then participates in this process [70,71]. This involves the action of N-sulfotransferases early in this process and 6-O and 3-O-sulfotransferase activity later, yielding HS sulfation motifs that finely regulate progression to the specified cell lineage [74]. HS-PGs regulate stem cells in niche environments in developmental tissues sequestering growth factors and maintaining stem cell viability [60,61,62,63,75].

### 2.2. CS Sulfation Motifs Expressed by Stem Cells

The 3B3[-], 7D4, and 4C3 CS sulfation motifs are expressed by the progenitor stem cells involved in articular cartilage development and are markers of tissue morphogenesis [60,61,63]. The HS chains of perlecan have important roles in the sequestration of growth factors within stem cell niches, thus regulating stem cell proliferation and viability and the attainment of pluripotency, i.e., yielding migratory stem cell populations that drive tissue development and promote tissue repair [30,76,77]. Perlecan is a hybrid CS/HS -PG in many tissues and in foetal human tissues has been shown to carry the 7D4 CS sulfation motif which has also been found in tissues undergoing morphogenetic transformation [78].

### 2.3. The Distribution of Sulfate Groups along GAG Chains Is Not Uniform

Numerous studies have demonstrated the non-uniform distribution of sulfate groups along HS chains as shown in Figure 1. Sulfate groups are also non-uniformly distributed along all GAGs other than HA which is the only non-sulfated GAG (Figure 2). The development of a number of mAbs to specific CS sulfation motifs has facilitated the mapping of these motifs along CS and DS chains [13,79,80,81]. Mapping of CS/DS GAG domains has also enabled the structural characterization of PGs [81,82,83]. A HS Interactome study documents a large number of HS interactive ligands [17]. An extensive range of HS modifications occur in tissues (Figure 2a). Sulfation patterns of CS and HS are variable in cancer (Figure 3a) and have been attributed to mutations in HS and CS modifying enzymes in a range of musculoskeletal disorders (Figure 3b).

### 2.4. The Glyco-Code of HS and the Enigmatic 3-O Sulfated HS Sulfation Motif

HS exhibits a linear helical form with unique small “kink(s)” possible where *D*-glucuronic acid is epimerized to *L*-iduronic acid due to the greater flexibility in glycosidic linkages around IdoA residues, which provides focal regions in the HS chain that can adopt a more compact conformation containing 3-*O* sulfation motifs [84]. 3-*O* sulfation in HS chains is generally rare, however, an abundance of HS3ST isoforms is present in tissues with varying substrate specificity making this functional structural modification of HS enigmatic. HS is a repository of the most diverse “sulfation codes” of any GAG [84]. This provides HS with the ability to selectively recognize and interact with specific proteins [85,86,87]. The diversity of proteins that HS can interact with is reflected in the proteins listed in HS and GAG Interactome databases. The HS sulfation code is not biosynthesized in a classic template-driven manner but arises from the action of tissue context-controlled spatiotemporal expression of a superfamily of *N*- or *O*-sulfotransferases [88,89]. Seven isoforms of 3-*O*-sulfotransferase (HS3ST-1, -2, -3_A_, -3_B_, 4, -5, and -6) introduce 3-*O* sulfation motifs into HS chains in the final step of HS biosynthesis and is generally rare in most tissues [90]. However, this is not always the case, as human follicular fluid HS has abundant 3-*O*-sulfation motifs [91]. Follicular fluid has critical roles to play in the fertilization of the oocyte and its nutrition in early embryonic development. The seven H3ST isoforms display significant differences in substrate specificities. HS3ST-1 preferentially generates the AT-binding HS sequence whereas HS3ST-3_B_ generates binding motifs for HSV-1 glycoprotein D [90]. The other HS3ST isoforms display overlap in the generation of AT or glycoprotein D-binding motifs, however clear differences in specificity among the seven HS3ST isoforms have been identified [92,93,94]. A number of proteins have been identified that preferentially bind to 3-*O*-sulfation motifs in HS including AT [95], FGFR [96], glycoprotein D [97], cyclophilin B [98], neuropilin-1 [99], HC II [62,63], tau glycoprotein [87] and SARS-CoV-2 spike glycoprotein [68]. HS sulfation sequences flanking the 3-*O* sulfation region provide specificity to these interactive 3-*O* HS sequences. Thus 3-*O*, 2-*O* and 6-*O* sulfation motifs in HS equip it with a diverse dynamic recognition system for the precise identification of bioactive cell regulatory molecules through a “sulfation code” [100,101]. This is reflected in the co-receptor functions of cell surface HS-PGs [68] and the diverse roles of HS-PGs in the stabilization and functional properties of ocular tissues [102], and neural activity in the CNS/PNS [103]. HS-PG interactions with neural cell surface cell attachment glycoproteins provide synaptic stabilization [104,105,106] and the signalling properties of neural networks [106]. The specific contributions of neurexin synaptic HS-PGs to neural stabilization and function [104] and the relatively recently identified dystroglycan interactive photoreceptor basement membrane HS-PGs Pikachurin [107,108,109] and Eyes shut (EYS) [110,111] also show the important tissue organizational functional properties that HS-PGs convey to specific tissues [24,44].

## 3. Analysis of Glycan Structure and Function

The enormous structural diversity of glycan substitution on proteins poses significant technical challenges that make it difficult to ascertain the specific functional roles of these diverse glycosylation patterns [112]. The reference manual Essentials of Glycobiology is an extremely useful reference source aiding in any technical issues (https://www.ncbi.nlm.nih.gov/books/NBK579918/ accessed 8 November 2022). Major advances in quantitative transcriptomics, proteomics and nuclease-based gene editing have opened the door on new ways to analyse the functional roles of glycosylation through targeting the expression of the biosynthetic enzymes responsible for these protein modifications. Furthermore, development of advanced software for in silico analyses of glycan-protein interactions are improving our understanding of the functional complexities of the glycoproteome [113]. Glycan microarrays have also proven to be extremely useful tools in deciphering glycan-protein interactions [114,115,116]. Artificial intelligence (AI) methods are also now being used in glycan structure-function studies and in drug design and are well suited to the analysis of large data sets [117,118,119,120]. A number of glycomic databases and glycomic tools are available to assist in such studies [118,119,120,121,122]. MatrixDB is a particularly useful database well served with current glycomics information and services [118,119]. The Kyoto Encyclopedia of Genes and Genomes (*KEGG*) [123,124,125] covers bioinformatics and data analysis in the areas of genomics, metagenomics, metabolomics, and omics studies dealing with molecular modelling, and simulation systems biology and translational research in drug design and developmental applications. KEGG provides Java graphics tools for browsing genomes and comparisons of genome maps and computational analyses of sequence and graphic data. *KEGG* is updated daily and available on *GLYCAN* (http://www.genome.jp/kegg/glycan/ accessed 9 November 2022) a database of carbohydrate structures has also been added to KEGG [126]. This includes information on glycan-related pathways; and a *Composite Structure Map* illustrating variations in carbohydrate structures within organisms.

The Gene Ontology (GO) resource provided by the Gene Ontology Consortium is the world’s most extensive source of information regarding the function of genes and gene products (http://geneontology.org accessed 18 November 2022) [127,128].

### 3.1. Advanced Analytical Techniques for the Examination of GAG Structure and Function and the Identification of Their Molecular Targets

#### 3.1.1. Neutron Scattering GAG Analysis

HS dynamically binds many signalling proteins such as growth factors, chemokines, and cytokines and regulates their activities. These binding interactions with proteins are affected by the geometry, flexibility, and rigidity of the HS chains and the conformations of their attached sulfate groups. Understanding these dynamic interactions at the molecular level provides fundamental insights into these cellular HS interactions and is informative in the development of compounds that can modulate these interactions. Neutron scattering is a powerful technique for the examination of ultra-fast molecular interactions of biological macromolecules such as GAGs and sulfation motifs are important interactive determinants which influence such interactions. Neutron spectroscopy allows the evaluation of dynamic GAG-mediated interactions at the picosecond (ps) to nanosecond (ns) level [129] and has found application in the evaluation of SARS-CoV-2 Papain-like Protease interactions with prospective therapeutic anti-viral agents [130].

#### 3.1.2. Electron Detachment Dissociation Analysis of GAGs

Mass spectrometry (MS) and tandem mass spectrometry are useful GAG structural analytic techniques that have been widely used. Collision-induced dissociation (CID) techniques have been developed for GAG fragmentation, producing glycosidic cleavages but do not yield a broad range of structurally informative data on GAG ring structures. Charge-transfer dissociation (CTD) and electron detachment dissociation (EDD) techniques were thus developed to fragment GAGs in a way that would aid analysis of gas-phase GAG ions to gain more comprehensive structural information [131]. CTD can be used with ion trap mass spectrometers to examine structural details of sites of HS modification, EDD and CTD both yield superior structural information than CID [132].

#### 3.1.3. Ion Mobility Mass Spectrometry Methods for Analysis of HS Structure and Sulphation

Tandem mass spectrometry using collision-induced dissociation or electron-based fragmentation is a well-established technique in GAG analysis but suffers from incomplete coverage of analytic GAG fragments they generate. IR and UV photofragmentation ion mobility spectrometry and ion spectroscopies offer more flexibility and greater coverage of molecular structure [133]. Full coverage of all structural species is important since variably sulfated HS species are master regulators of biological processes. For example, 3-O-sulfated HS/heparin has crucial anticoagulant activity, however, therapeutic applications have been hampered by a lack of sufficiently discriminative tools to fully decipher HS structure-function relationships. The use of synthetic 3-O-sulfated standards, comprehensive HS disaccharide profiling, and cell engineering has addressed this deficiency demonstrating previously unappreciated differences in 3-O-sulfation profiles in clinical heparins and HS3ST generated structural variation in cell surface HS [134]. This has facilitated the correlation of functional differences in anticoagulant activity and PF4-induced thrombocytopenia side effects in therapeutic heparins. Cells expressing HS3ST4 isoenzyme generate HS with potent anticoagulant activity but weak PF4 binding providing insights into 3-O-sulfate structure-function in new generation tailored cell-based heparins [134]. Elucidation of HS/Heparin backbone biodiversity is a major challenge. Cryogenic infrared spectroscopy of mass-selected ions can distinguish isomeric HS tetrasaccharides differing only in the configuration of their hexuronic acid building blocks [135]. High-resolution infrared spectral analysis of a systematic set of synthetic HS stereoisomers has revealed their characteristic spectral features demonstrating the technique’s potential for sequencing of HS and structurally related biomolecules [135].

#### 3.1.4. X-ray Crystallography

Leukocyte common antigen-related (LAR) protein is a type IIa receptor tyrosine phosphatase (RPTP) which has roles in signal transduction and the regulation of axonal growth and regeneration. HS and CS regulate LAR signalling but lead to differing regulatory outcomes. X-ray crystallography provides important insight into type IIa RPTP interactions with CS and HS. A greater understanding of such interactions may assist in the design of novel therapeutic compounds to regulate nerve repair [136].

#### 3.1.5. In Silico Molecular Docking Simulations

The development of advanced computer software has facilitated molecular simulations of docking interactions between GAGs and protein interactive structures. This has been used to examine the interaction between SARS-CoV-2 Spike protein and cell surface HS on prospective host cells to better understand the COVID-19 infective process [137]. Similar interactions between the roundabout (Robo) Slit receptor [138], HS and Slit nerve guidance proteins have also been evaluated using molecular docking [139]. The Robo transmembrane receptors interact with Slit nerve guidance proteins to regulate axonal growth during neural development, Robo-Slit interactions are modulated by HS. Molecular docking studies of heparin oligosaccharides with the Hep-II heparin-binding domain of fibronectin have demonstrated the importance of the conformation of sulfate groups in this interactive process [140]. Heparin/HS binding sites on αvβ3 integrin have also been mapped using molecular docking methodology [141]. Molecular modelling of HS interactions with growth factors has provided invaluable information on how to optimize the activity of cellular growth factors [142].

#### 3.1.6. Application of Artificial Intelligence (AI) in GAG Analysis

AI methodology has been successfully applied to the analysis of GAG structure [143]. Machine learning algorithms have the potential to perform high throughput pattern recognition analyses on enormous glycomic datasets, extracting meaningful information from complex or noisy data for both descriptive and predictive purposes, and offering significant diagnostic, prognostic, and therapeutic value [144]. AI and deep machine learning have been applied to the analysis of large glycomic datasets in drug development [145,146,147,148]. This methodology has also been used to analyse glycosylation patterns and the stereoselectively of carbohydrates [149,150] and to make predictions on the contributions that atomistic environments make to glycan selectivity. AI shows considerable promise in the analysis of glycomic databases and offers significant potential in the improvement of our understanding of glycan ligand interactions which may be of therapeutic application [144]. Increasing amounts of glycomics data are being generated through advanced glycoanalytics technologies. Many glycomics databases and tools for glycomics manipulation are now available. AI provides a powerful analytic tool for analysis of large data sets [151]. AI methods integrated with prediction software in glycoinformatics approaches are emerging to provide further improvement in glycomics analysis [117].

#### 3.1.7. Analysis of GAG 3D Structure

The enormous structural diversity of GAGs and PGs has hampered GAG analytics by standard methodologies. Advanced computational software applied in molecular modelling, docking dynamic simulations and measurement of binding kinetics are proving useful in the analysis of 3D GAG interactions with their prospective molecular targets [121]. Significant inroads have been made in the integration of data generated by computational scientists and glycobiologists to improve our understanding of the considerable complexities of the instructive glycan world and its cellular interactions [121]. GAGs are complex conformationally diverse molecules. GAG-DB is a curated database of the 3D features of CS, DS, HS, HA and KS and their oligosaccharides complexed with proteins [122]. The entries are covered by X-ray fibre diffractometry, solution NMR spectroscopy, and scattering data associated with molecular modelling. This database is designed with navigation tools to query the database interfaced with Protein Data Bank (PDB), UniProtKB, and GlyTouCan (universal glycan repository) identifiers. GAG-DB provides detailed information on GAGs and protein interactive ligands using several open access applications and is freely available [121], as is an interface application GAG Builder [122]. GLYCAM-Web GAG Builder, (www.glycam.org/gag accessed 18 November 2022) is an additional web tool designed for the non-expert for rapid and straightforward prediction of 3D structural GAG models [152]. The user is provided with coordinate visualisation files where counter ions and water may also be added as desired. Heparin/HS, CS, DS, KS, and HA are all supported and analysis of sulfation patterns in novel sequences.

#### 3.1.8. Advanced High Precision Quantum Cellular Imaging Methodology for the Identification and Quantitation of Single Molecule Contributions to Cellular Phenotypes

Significant advances in cell imaging at the sub-nanometre level for single molecule analysis using Quantum imaging techniques [153] have provided important advances in the assessment of cellular changes in disease processes and the roles of specific proteins [154,155]. Peptide painting of single molecules with quantum dot semi-conductor nanoprobes [156,157] has enabled the examination of specific contributions single molecules make to cellular phenotypes [155,158,159]. Quantum dots are fluorescent nanoparticles with narrow-band, size-tuneable, and long-lasting emission profiles [160], which have proved useful in molecular imaging in cancer medicine [161]. This methodology has also been applied to live cell imaging and is capable of automation [162,163]. Thus, it may prove useful in the spatio-temporal assessment of HS modifications [164] occurring during tissue morphogenesis and repair [165,166]. Quantum dot-based nanoprobes developed for AT and FGF-2 have been used to map the cell surface distribution of the rare 3-*O* HS sulfation motif [167]. Integration of deep machine learning and unbiased automated high-content screening procedures of robotically controlled high throughput cell cultures has been applied in the identification of complex disease signatures [168]. The high precision provided by the aforementioned methodologies may help decipher the molecular complexities involved in spatio-temporal and conformational HS modifications known to occur in tissue morphogenesis and ECM remodelling in disease processes [169]. Deciphering the complexity of ECM glycosylation signatures will be invaluable in the discovery of novel therapeutics and biomarkers in the diagnosis and treatment of diseases [151,170,171,172]. The HS interactive sequences with Wnt, lipoprotein lipase, AT, FGF-2 and FGFR have been determined (Figure 4). A structure-function computational study has investigated how the rare *O*-3 HS sulfation presentation contributes to protein binding specificity [173].

#### 3.1.9. GAGome Biomarkers of Disease

While GAGs have established roles in normal physiology and disease, understanding the biological functions of specific GAG structures is difficult because of their great structural heterogeneity. Urine and plasma GAG profiles (GAGomes) [174] nevertheless have been useful as tumour metabolic biomarkers for early cancer detection [175].

#### 3.1.10. Transcriptomics and Its Application in the Analysis of Glycoproteomics

Glycosylation is an extremely diverse post-translational eukaryotic cell-mediated modification that occurs in proteins and involves complex metabolic and glycosylation pathways that orchestrate proteomic amplification of PG biodiversity [113]. In silico approaches with predictive capability in the analysis of cellular glycosylation outcomes are emerging, and revolutionizing glycosylation maps and genetic approaches that address the modification of glycoproteomic functional properties [112,176,177]. Transcriptomics, genetic screening, and engineering of glycosylation patterns at the single-cell level are significantly advancing glycan mimetics and the design of therapeutic glycoproteomic modulators of cellular behaviour which are being applied in the treatment of human disease [178,179,180,181]. Single-cell transcriptomics and the increased precision it provides are revolutionizing the analysis of variations in the structure of HS-PGs and how these contribute to spatio-temporal developmental processes. Transcriptomics analysis of ovarian tissues has also shown that HS/heparin and CS/DS GAG biosynthesis occurs by two pathways with the biosynthesis of HS- and CS-PGs differentially regulated. HS-PGs and CS-PGs are significantly upregulated in folliculogenesis. Ovarian GAGs promote gonadotropin-induced follicle development in the mouse acting through stimulation of angiogenesis consistent with roles assigned to perlecan in the development of a number of these tissues [182,183]. Transcriptomics has also been used to examine the spatiotemporal diversity and regulation of GAG biosynthesis in normal homeostatic tissues and in those undergoing disease processes [24].

#### 3.1.11. Antibodies That Detect HS Sulfation Motifs

Sulfation motif antibodies to HS and CS have been used to delineate the topographical distribution of these entities in tissues undergoing tissue morphogenesis [13,184,185] related to cell behaviour [186,187] and structure-function inter-relationships [188,189,190,191] and the bioactivity of HS and CS in normal and pathological tissues [192,193,194,195,196,197,198,199,200] and the modifications in HS and CS that occur in tissues [13,24,201,202,203,204]. Glycan microarrays provide important information defining the specificities of these antibodies. Methods have been developed with GAG microarrays and computational modelling to analyse GAG-protein interactions [205].

#### 3.1.12. Genome-Wide Analysis of Highly Specific CRISPR/Cas9 Gene Editing Sites in Musculoskeletal Disease and GAG and PG Systems

Clustered regularly interspersed short palindromic repeat (CRISPR) technology employing the RNA-guided nuclease Cas9 has rapidly dominated the genome engineering field as a unique and powerful gene editing tool. CRISPR/Cas9 gene editing technology is being increasingly used to interfere with GAG biosynthesis in model systems to better understand the roles of GAGs in specific musculoskeletal conditions. CRISPR/Cas9 has been used to generate aggrecan deficient stable chondrocyte cell lines [206] and applied in studies on mucopolysaccharidosis I to examine how GAGs impact musculoskeletal tissues in this condition [207]. CRISPR/Cas9 will undoubtedly see increased application in studies aimed at understanding how GAGs function in musculoskeletal tissues in health and disease. Transcriptomics have been widely applied to examine spatio-temporal GAG changes in diseased tissues, in the identification of new basement membrane HS-PGs and to assess how PG profiles affect tissue architecture in disease [208,209].

**Figure 4 ijms-24-01148-f004:**
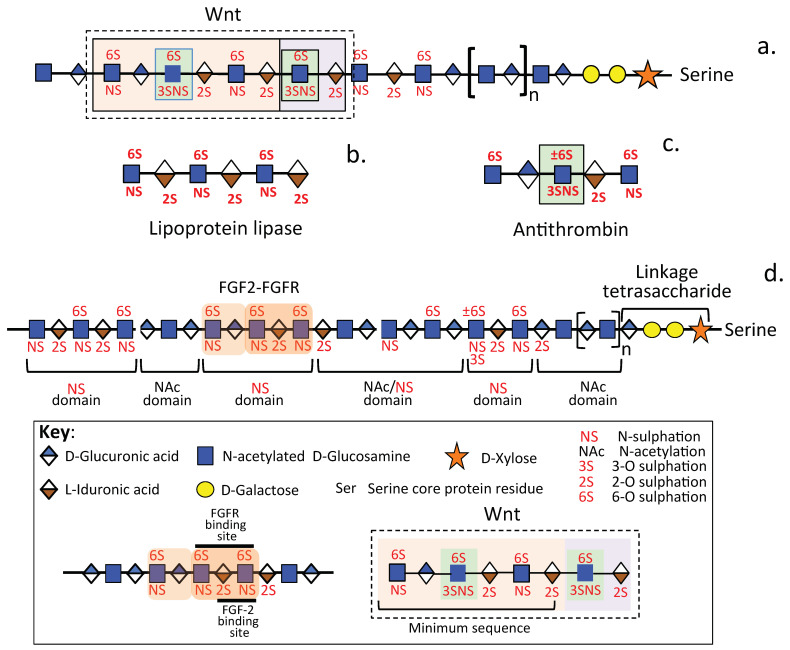
Identification of specific ligand interactive glyco-sequences in HS. The structures shown are those proposed by Gao et al. [188] (**a**) Cummings [210] (**b**,**c**) and Gomes et al. [211]; Wang et al. [212] (**d**). The orange highlighted area in HS in (**d**) depicts FGF-2 and FGF-R binding sites however flanking structures also contribute to specificity in a functional tetra or pentasaccharide arrangement as determined by NMR spectroscopy [213].

## 4. HS-Proteoglycans

### 4.1. HS-PGs Are Broadly Distributed ECM, Cell Surface, Intracellular, and Nuclear Components (Table 1)

HS-PGs are multifunctional cell regulatory molecules that have a widespread distribution in tissues and occur as the cell membrane, intracellular and ECM PGs (Table 1). They are found localised around mesenchymal stem cell niches and have roles in stem cell differentiation and production of defined progenitor cell lines with roles in tissue development, ECM remodelling and tissue repair [60,61,63]. Assessment of the GO (Gene Ontology) biological process categories and the KEGG (Kyoto Encyclopedia of Genes and Genomes) pathways enriched in heparin/HS interactome members demonstrates the diverse interactive properties and biological processes HS-PGs are involved in [17,214].
ijms-24-01148-t001_Table 1Table 1Extracellular, Cell membrane, Intracellular and Nuclear HS-Proteoglycans.LocalisationProteoglycanReference
Type XV Collagen[215,216]
Type XVIII Collagen[217,218]ECMAgrin[219,220,221,222,223,224,225,226]
Perlecan[77,227]
SPOCK1, 2 (Testican-1, 2)[228]
Glypican 1–6[229,230,231,232]
Syndecan 1–4[233,234,235,236,237,238]Cell surfaceBetaglycan (TGF-β receptor III)[239,240]
CD44E, Epican[241,242,243,244,245,246]
CD47[247,248]
NRP-1[249,250]IntracellularSerglycin[251,252,253]
Neurexins α, β, γ[104,106,254]NuclearPerlecan, Syndecan-1–3, Glypican-1[255,256,257,258,259]


### 4.2. Betaglycan

Betaglycan is a multifunctional 250–280 kDa transmembrane HS/CS co-receptor for the TGF-β superfamily [239,260], it forms a functional homodimer at the cell surface that contains inhibin, FGF-2, Wnt and TGF-β binding sites [240,261,262,263,264,265,266,267,268]. The HS chains of betaglycan bind FGF-2 and but are not required for the binding of TGF-β ligands. Wnt also binds specific HS sequences independently of the TGF-β binding activity of betaglycan [269]. HS inhibits Wnt signalling while CS promotes Wnt signalling [269,270]. Betaglycan’s N-and O- linked oligosaccharides also influence betaglycan’s ligand binding activity modulating its growth factor-mediated, vascular and cancer cell migratory properties [271] and interactions with inhibin A and B [272]. Fragments of betaglycan released from the cell surface by plasmin and MMPs act as circulating antagonists to cell-based betaglycan activity. Co-localization of inhibin/activin protein subunits and betaglycan has been observed in the human brain [273,274]. FGF-2 carried by betaglycan stimulates neural proliferation and differentiation [275]. TGF-β enhances adult neurogenesis in the sub-ventricular zone and has anti-inflammatory properties in the adult brain [276,277]. Activin and inhibin regulate the secretion of follicle-stimulating hormone (FSH), affecting the differentiation, proliferation, and cellular function of many cell types [278]. Activin receptors are highly expressed by neuronal cells and their levels are further elevated following brain trauma, hypoxic/ischemic injury, mechanical irritation, or chemical damage [279]. FGF-2 is neurogenic and has neuroprotective anti-apoptotic properties linked to the promotion of brain lesion-induced upregulation of activin A activity [279,280,281].

### 4.3. CD47

CD47 is a HS/CS hybrid transmembrane PG multifunctional receptor of haematopoietic cells. HS substitution on CD47 inhibits T cell receptor signalling by thrombospondin-1 (TSP-1). CD47 is a TSP-1 receptor for TSP-1 and regulates cellular migration, proliferation, and survival of vascular cells in innate and adaptive immune regulation [282,283]. TSP-1 acts via CD47 to inhibit nitric oxide (NO) signalling in the vascular system and supports blood pressure by limiting endothelial nitric oxide synthase (eNOS) activation and endothelial-dependent vasorelaxation [284]. CD47 is a ligand for SIRPα (signal regulatory protein α), with tyrosine-based activation motifs or CD172a [285,286,287]. This has been termed an innate immune checkpoint sending a “do not eat me” signal to macrophages [288] allowing tumour cells to avoid detection and clearance. Blockade of the CD47-SIRPα interaction using humanised antibodies to CD47 (Hu5F9-G4) has yielded promising preclinical results in a number of human malignancies including paediatric brain tumours: medulloblastoma, atypical teratoid rhabdoid tumours, primitive neuroectodermal tumour, paediatric glioblastoma, and diffuse intrinsic pontine glioma [289]. By targeting the CD47-SIRPα immunological checkpoint glioblastoma development can be inhibited and the activity of phagocytic, dendritic and T-lymphocytes enhanced promoting clearance of tumour cell in innate and adaptive immunity [290,291,292,293]. Targeting of TSP-1:CD47 interactions is a promising anti-thrombotic approach, engineered CD47 cyclic peptide TAX2 (CEVSQLLKGDAC) selectively inhibits TSP-1:CD47 interaction, decreasing platelet aggregation and platelet-collagen interactions under arterial shear conditions and not associated with increased bleeding risk. TAX2 is an innovative antithrombotic agent and a novel antagonist of TSP-1: CD47 interactions [294].

### 4.4. Neuropilin

Neuropilin-1 (NRP-1) is a multidomain cell surface membrane protein with roles in angiogenesis and neural development [295]. NRP-1 contains a PDZ cytoplasmic domain with a limited ability to transduce signals across the cell membrane and relies on co-operation with other cellular receptors to regulate cell signalling [296,297]. NRP-1 is shed from cell surfaces as a soluble isoform that interacts with other membrane receptors, their ligands and HS. NRP-1 binds highly sulfated and completely de-sulfated regions of heparin but has a preference for 3-O sulfated HS [99] or chemically modified heparins bearing one or two sulfate groups, e.g., it binds heparin with a single 6-O sulfate group better than heparin with any two N-sulfate, 6-O sulfate and 2-O sulfate groups [298]. The b1 domain, a1, domains and the L2 linker region of NRP-1 all make contributions to interactions with heparin and HS [295]. NRP-1′s preferential binding to extended sulfated glycan structures indicates that it can potentially bind large segments of HS chains.

### 4.5. Neurexins

The shape and properties of a synapse are a function of its molecular organization [299,300,301,302]. The neurexins are a family of pre-synaptic HS-PGs which interact with several post-synaptic adhesion molecules in the synaptic cleft. Neurexin isoforms exist as PGs with variably sized core proteins containing a single HS chain. The GAG sequence of this HS chain may differ between neuronal cell types, conferring specificity to neurexin interactions with cell adhesion molecules [104,303]. Several alternatively spliced isoforms of neurexin core proteins have also been identified. These contain laminin, neurexin, sex-hormone-binding (LNS) domains interspersed with EGF repeat domains in some cases. α-Neurexin core protein isoforms contain six LNS domains while β-neurexins contain a single LNS domain and no EGF domains. The neurexins are transmembrane cell surface HS-PGs which attach to the pre-synaptic cell membrane through PDZ domains. Presynaptic neurexins and type IIa protein tyrosine phosphatases (RPTPs) interact with a range of postsynaptic ligands in functional synaptic organization. Leucine-rich-repeat transmembrane neuronal proteins (LRRTMs) initially interact with neurexin core protein domains however it is the HS side chain neurexin interactions with LRRTMs that is critical to the induction of presynaptic differentiation and synaptogenesis [303]. Neurexin-neuroligin interactions also mediate synapse development and function, incorrect interactions between these occur in autism and schizophrenia [106]. The HS chain of neurexins binds postsynaptic neuroligins and LRRTMs, which complement neurexin core protein-mediated interactions [303]. Variation in the fine structure of these HS chains aids in fine-tuning synaptic function in brain development [304]. The dynamics of synaptic organisation controls diverse interactions with trans-synaptic signalling molecules. Alternatively spliced forms of the neurexins have central roles to play in this dynamic code [305]. Presynaptic neurexins regulate synaptic interactions through differential binding to a range of postsynaptic ligands, modulating the input/output signals generated in the synaptic cleft. Neuroligins are major post-synaptic adhesion molecules that interact with the neurexins, these occur in a very diverse range of alternatively spliced forms offering a large range of synaptic regulatory signals [304]. Mutations in genes encoding the neurexins and their ligands have been observed in neuropsychiatric disorders such as schizophrenia, autism, epilepsy, and Tourette syndrome, showing the neurexins have central roles to play in the control of synaptic plasticity [306,307,308,309,310,311,312,313,314,315,316]. HS-PGs have critical roles to play in the development of Alzheimer’s disease (AD) [317].

### 4.6. Pikachurin

Pikachurin, (AGRINL and EGF-like, fibronectin type-III and laminin G-like domain-containing protein), is encoded in humans by the EGFLAM gene. Pikachurin is a 110 kDa dystroglycan (DG)-interactive protein with roles in the coordination of precise associations between the photoreceptor ribbon synapse and bipolar dendrites [109]. The binding of pikachurin with DG requires DG glycosylation and divalent cations. Incorrect interaction between pikachurin and DG has been noted in muscular dystrophies often associated with eye abnormalities [318,319]. Pikachurin has been reported to be substituted with HS and CS although the precise attachment points to the core protein have yet to be published [44]. The pikachurin core protein contains multiple fibronectin type III repeat, EGF and LamG modules which are known to be protein interactive domains [320].

### 4.7. Eyes Shut

Eyes shut *(Eys)* encodes a predicted retinal basement membrane modular 250–350 kDa PG closely related to agrin and perlecan with roles in photoreceptor organisation and ECM remodelling in the developing retinal epithelium [111]. Like agrin and perlecan, *Eys* contains multiple EGF and LamG modules and a central Serine-Threonine-rich module containing multiple putative GAG attachment sites. Ornate immunolocalisations of *Eys* have been undertaken demonstrating its spatio-temporal localization and apparent roles in the development of the insect retinal epithelium. *Eys* is secreted by photoreceptor cells in *Drosophila* however in the honey bee, *Eys* is not secreted by photoreceptor cells demonstrating it has evolved variable roles in eye development [110]. The normal development and function of visual photoreceptors are essential for mammalian eye health and visual acuity. Mutations in genes encoding proteins involved in photoreceptor development and function produce a range of inherited retinal dystrophies [321]. Mutations in the *EYS* gene are a common cause of autosomal recessive retinitis pigmentosa (arRP) in Chinese and Japanese sub-populations [322], yet the role of EYS PG in humans remains to be fully determined [110,323]. In photoreceptor cells, EYS may be involved in the stabilization of the ciliary axoneme in both rods and cones however the occurrence of multiple EYS isoforms suggests additional roles may yet be uncovered.

Figure 5 and Figure 6 are presented in this review to show the diversity in structure of the HS-PGs but also to illustrate functional elements in common across HS-PG members. There have been many excellent reviews on all aspects of HS-proteoglycan biology and the interested reader is recommended the following studies for further information [29,30,53,76,77,80,228,230,231,324,325,326,327,328,329,330,331,332,333,334]. Our discussion on HS-PGs will focus mainly on the more recent members of the HS-PG family, namely the neurexins, eyes-shut and pikachurin rather than recapitulating information on the other members of this PG family which are already widely available.

## 5. Function Defining Properties of HS-PGs and CS-PGs in Neural Tissues

Cerebroglycan (GPC-2) is a developmentally regulated GPI anchored integral membrane HS-PG found exclusively in the developing nervous system [335,336]. The developmental expression pattern of cerebroglycan and other glypicans indicates they have instructive roles in the regulation of neuronal growth and axonal guidance during the formation of neural networks [337]. Glypican HS-PG family members have interactive roles with the neural guidance proteins netrin and slit and robo roundabout receptors to coordinate embryonic neural network development [338,339,340,341,342]. A specialized form of notochordal aggrecan, a CS-KS lectican PG member containing the human natural killer-1 (HNK-1) glycan epitope also has instructive guidance roles over neural crest precursor cells during these developmental processes [49]. Formation of neural tissues and neural networks is a highly coordinated process with essential roles for CS-PGs and HS-PGs ensuring that individual neurons precisely reach their designated targets in neural networks during tissue development [49]. CS-PGs and HS-PGs also have roles in neuronal proliferation, differentiation [343], synaptic development and specificity determination [106,334,344] defining tissue function in adult neural tissues in health and disease [15,47,54,345,346,347,348].

### 5.1. The Synapse and Perineuronal Nets (PNNs) Are Highly Sophisticated Sensory Glycocalyx-like Structures of Functional Significance in Neural Homeostasis and Neurodegenerative Pathophysiology

Neural tissues are quite unlike any other tissue in the human body. The neural ECM is dominated by GAGs and is devoid of the fibrillar collagenous and elastic networks that are prominent stabilizing structures in connective tissues. Neural PGs have roles both in tissue stabilization and also provide instructive cues to neural cell populations in specialized niche structures and ionic microenvironments that ensure cellular viability and optimal cellular activity. CS-PGs and HS-PGs both have important functional properties in these tissues and coordination of their interactive properties is a prominent feature of neural tissue function and homeostasis [348]. Thus, while HS-PGs are pigeonholed with important intrinsic instructional properties from in vitro studies these are not enacted in isolation in vivo but occur in environments containing a myriad of influential modulatory entities in a cell-mediated interplay of these stimulatory factors. The pleiotropic nature of HS amply equips HS-PGs with properties to excel in such a mixed environment [6,30,77,349].

### 5.2. Roles for HSPGs in the Neuronal Synapse and PNNs

HS-PGs and CS-PGs have co-ordinated cell instructive roles in neuronal synapses and in PNNs regulating tissue homeostasis and tissue remodelling in health and disease. PNNs are condensed forms of brain ECM that surround the soma and proximal dendrites of subsets of neurons, enwrapping and protecting synaptic terminals [350]. PNNs are remarkably dynamic structures undergoing changes that influence cognition and memory processes [351]. Experimental disruption of PNNs in adult animals reactivates critical period plasticity and under certain conditions may actually improve memory. During these so-called “critical periods “the nervous system is especially sensitive to environmental stimuli, and this represents an important aspect of learning and memory processes [352]. During this critical period neuronal circuits undergo dynamic remodelling, however, if for some reason, an appropriate stimulus is not received during this “critical period” then this opportunity to learn a potential new skill may not occur. HS-PGs and CS-PGs both have established cell instructive roles in the coordination of axonal migration in complex environments during embryonic neural network formation [49]. While HS-PGs are associated with PNNs, they are not however considered to be integral functional components of the PNN scaffold which is composed primarily of CS-PGs. The sulfation patterns of the PNN CS-PGs are however recognized to have important cell instructional roles, a feature shared by HS-PGs [103,351]. Neurons in the PNNs also synthesise cell surface HS-PGs through which instructive ECM cues to cells may be enacted upon. Such ECM cues have instructive directional roles over correct neuronal functional properties, disruptions in such interactions can contribute to the development of neurological disorders [353]. Proteomic analysis of schizophrenia and AD neural tissues has demonstrated the many proteins involved in such disruptions in normal synaptic activity [354].

Neurexins (Nrxn 1, 2, 3) are highly interactive specialized HS-PGs which stabilize the synapse through an extensive range of interactive ligands that have critical roles in correct neural function [355,356]. From an evolutionary standpoint, HS is a glycocalyx molecule that developed through strict evolutionary selection pressure to become a molecule with properties of molecular recognition and information storage and transfer to regulate cellular behaviour. Thus, in the present day, the synapse might be considered a highly specialized glycocalyx where the HS side chains of the neurexins have similar regulatory functions. Dysfunctional neurexin activity has been identified in a range of psychiatric disorders including schizophrenia, autism, intellectual disability, and addictive disorders [357]. The variable HS sulfation patterns of neurexins contributes to their diverse interactive features in the synapse with a wide range of ligands in health and disease [103,358]. The synapse contains a very diverse range of HS interactive ligands that are important for neural activity and message transduction in neural networks [106,344].

Recently, attention has also focused on the role of PNNs in the control of neuronal and synaptic plasticity in health and disease [359,360], it is now clear that PNNs also have important roles to play in cognitive learning processes and the modulation of memory. Disruption in PNNs has been observed in AD [361,362,363,364] and with increased levels of Tau pathology in neurodegenerative conditions [365]. HS-PGs also have roles in the deposition of pathological protein aggregates in brain tissues [317]. PNNs are extensively lost in AD in proportion to the level of amyloid plaque deposition and Tau hyperphosphorylation, this loss of PNNs appears to be mediated by glial cells [366] and by dysregulation of astrocytes and astrocyte-neuronal communication [367,368]. HS-PGs mediate some of the effects of astrocytes on synaptic function and participate in the astrocyte-mediated brain injury response [369]. PNNs, neuronal cell surface receptors and interactive molecules in the ECM have roles in the pathophysiology of a number of psychiatric disorders including schizophrenia, autism and mood disorders, AD, and epilepsy [359,360,370,371]. Such interactions can potentially affect neuronal pre- and postsynaptic terminals [353,354,358]. Glial and astrocyte activation in neurodegenerative conditions may also affect crosstalk with neuronal cell populations that may modulate synaptic functions and plasticity leading to synaptic dysfunction. Furthermore, the loss of PNNs in AD and the protection they normally provide may expose neuronal cell populations to toxic components generated by neuro inflammatory processes during the pathogenesis of neurological disorders contributing to mitochondrial dysfunction and apoptosis of neurons [362,372].

The GPI-anchored glypican HS-PG family, have critical roles in the development and function of synapses and along with alternatively spliced members of the neurexin HS-PG family form synapse-organizing protein complexes and also serve as ligands for leucine-rich repeat transmembrane neuronal proteins (LRRTMs) and members of the leukocyte common antigen-related receptor protein tyrosine phosphatase (RPTP) family [351,371]. Neuronal RPTP-σ acts as a PG receptor binding HS-PGs and CS-PGs and these convey cell instructive signals. CS-PGs generally transmit inhibitory signals that down regulate proliferation and neural outgrowth from gliotic scars impeding functional repair of brain tissues from traumatic injury whereas HS-PGs convey signals that promote neural proliferation, migration, and neural outgrowth [49,103]. The collective interplay of HS-PGs and CS-PGs thus regulates tissue development, homeostasis, tissue remodelling and repair processes in health and disease [366,373,374].

### 5.3. Cell Regulatory Roles of Dually Modified HS/CS PG Co-Receptors

Dual modified HS/CS cell surface PG co-receptors are multifunctional, modulating Wnt, ShH, TGF-β and FGF signalling in development and disease. Incorrect cell signalling responses to secreted growth factors are linked to the development of a number of diseases [269,270,375,376]. Dual modification of co-receptors with HS/CS facilitates simultaneous binding of growth factors from different growth factor families modulating multiple regulatory processes. HS and CS promote opposing cell instructive processes in many cases and may represent a molecular “on-off” switch. For example, HS inhibits Wnt signalling through betaglycan and CD47-mediated interactions whereas CS promotes Wnt signalling. CS and HS also have opposing effects on neuronal proliferation, migration, and outgrowth through RPTP-σ interactions with CS providing inhibitory signals. Cell surface hybrid CS/HS PG co-receptors include members of the syndecan family, betaglycan and the TSP-1 receptor CD47. A substantial proportion of NRP1 is a PG modified with either HS or CS on a single conserved Ser residue in the core protein. GAG-free forms of NRP-1 are also common. The composition of the NRP1 GAG chains differs between endothelial cells and smooth muscle cells however a single GAG chain cannot contain both HS and CS simultaneously, thus endogenous NRP1 exists as either an HS-PG or CS-PG but not as a hybrid PG. The Ser residue that is substituted with GAG is located between the b1b2 and MAM domains of NRP1 and is conserved between vertebrates. NRP2, a mammalian homolog of NRP1, does not have this conserved Ser residue [250]. NRP-1 and 2 class III semaphorin receptors interact with semaphorin and VEGFA in axonal guidance in embryonic neural network formation. In adult tissues NRP-1, 2 are pro-angiogenic stabilizing VEGF/VEGFR interactions that promote tissue development and repair, tumour growth, migration, and invasion [250]. Two NRP complement-binding homology domains (a1 and a2), are essential for binding to Sema 3 and Plexin A during axonal guidance. NRP b1 and b2 coagulation factor V/VIII homology domains mediate VEGF binding. NRP-1 binds highly sulfated and completely desulfated heparin oligosaccharide regions but has a preference for 3-O sulfated HS [99]. Cell surface betaglycan homodimer contains inhibin, FGF-2, Wnt and TGF-β binding sites [240,261,262,263,264,265,266,267,268] and co-localizes with inhibin/activin protein subunits in the brain [273,274]. CD47 is another hybrid CS/HS neuron co-receptor associated with neurodegenerative disorders. CD47 ligands include SIRPα, TSP-1 and integrins. These regulate cell adhesion, proliferation, apoptosis, cell migration, homeostasis, and immune function in neural tissues. CD47-SIRPα interactions modulate T cell activation in innate immunity. [285,286,287]. TSP-1 acts via CD47 to inhibit vascular NO signalling regulating blood pressure by limiting eNOS activation and endothelial cell-dependent vasorelaxation. CD47 interaction with SIRPα regulates innate self-recognition [282]. CD47 TSP-1 interactions regulate vascular remodelling under stressful conditions such as brain trauma and neurological disease [377,378]. CD47 protects synapses from excess microglia-mediated remodelling during tissue development and trauma [379] and has neuroprotective roles in neuroinflammatory processes occurring during pathological tissue remodelling in neurodegenerative disorders that reduce cognitive processes and memory [380,381].

In textbooks, perlecan is often described as a HS-PG, referring to vascular perlecan produced by endothelial cells which is a mono-substituted HS-PG. However, many cell types (chondrocytes, IVD cells, meniscal cells, smooth muscle cells) produce perlecan where at least one of its HS chains is replaced by CS, thus perlecan in many tissues is present as a HS/CS hybrid PG. Few studies have examined the effect of this replacement of HS by CS on the functional biology of perlecan. One study has shown the CS chain of perlecan regulates collagen fibrillogenesis [382]. Another study showed CS regulated the release of FGF-2 sequestered by the HS chains, with the release of HS only occurring after the removal of CS by bacterial chondroitinases, however, the physiological enzyme responsible for this was not known at this time [383]. The discovery of hyaluronidase-4 as a CS hydrolase is a mammalian enzyme capable of removing CS from perlecan [384]. The HS editing enzymes heparanase, Sulf-1 and 2 also remove 6-O sulfated HS from HS-PGs [385,386] and this regulates angiogenic responses of endothelial cells to FGF-2 and VEGF [387]. Co-operation between HS and CS chains in syndecan are also involved in the improved binding and transfer of midkine and pleiotropin to their cognate receptors [388]. Thus, the dual modification of perlecan with HS and CS may further improve its growth factor binding capabilities.

### 5.4. The Roles of LamG and EGF-like Core Protein Motifs in HS-Proteoglycans

Agrin, laminin, perlecan, neurexin and pikachurin interact with α-DG through their LamG domains [109,389,390,391]. LamG domains are also prominent in agrin, perlecan and collagen XVIII core proteins. Pikachurin is essential in the maintenance of the photoreceptor ribbon synapse that interacts with dendritic processes of bipolar neurons [109]. Eys, a basement membrane HS-PG also interacts with matriglycan and α-DG through its LamG domains to stabilize the photoreceptor basement membrane [392]. Drosophila Eys contains 4 N terminal LamG domains, human Eys and Zebrafish Eys homologs contain 5 LamG domains. EGF, calcium-binding EGF (cb EGF) and EGF-like domains are also prominent components of the Eys core protein and other HS-PGs. Drosophila Eys contains 10 EGF and 4 cbEGF motifs, the human Eys homolog has 18 EGF, 3 EGF-like and 7 cbEGF domains, Zebrafish Dys has 20 EGF, 1 EGF-like and 18 cbEGF domains. EGF-like domains have previously been identified in the globular domains of members of the lectican CS-PG family and shown to promote cell proliferation and migration in some cancers [393,394,395] however the specific roles of the EGF-like domains in the HS-PGs have not been ascertained, it is not known if these also contribute to cell proliferation and migration promoted by HS-PGs.

α-Nrxn-1, 2, 3 in the presynaptic cell surface are essential for neurotransmission and linked to neuro-developmental disorders such as autism or schizophrenia. Diverse interactive partners of α-Nrxn depend on alternative splicing, and neuroligins (Nlgn) and αDG [356]. The trans-synaptic complex with Nlgn1 contributes to α-Nrxn function. Interactions of α-Nrxn with αDG, neurexophilins (Nxph1) and Nlgn2, ligands occur specifically at inhibitory synapses, however, these interactions are incompletely understood [356,396]. The binding epitopes of αDG and Nxph1 on Nrxn1α show binding is mutually exclusive. An unusual cysteine bridge and complex glycans in Nxph1 ensure binding to the second laminin/neurexin/sex hormone binding (LNS2) LamG domain of Nrxn1α, this association does not interfere with Nlgn binding at LNS6 [397,398,399,400]. αDG, in contrast, interacts with both LNS2 and LNS6 mostly via LARGE (like-acetylglucosaminyltransferase) generated glycans. The binding of αDG at LNS2 prevents interaction of Nlgn at LNS6 by steric hindrance. Expression of αDG and Nxph1 and alternatively spliced Nrxn1α may prevent or facilitate the formation of distinct trans-synaptic Nrxn·Nlgn complexes, contributing to diverse synaptic interconnections [401,402].

α-DG is a highly glycosylated basement membrane receptor component of matriglycan, a novel extracellular glycoprotein [403]. Multiple glycosyltransferases have key roles in function-defining modifications to α-DG [404,405,406]. A functional *O*-mannose structure uses phospho-ribitol phosphotrisaccharide bridges to link to 30 disaccharides of α-DG that bind ECM proteins [407]. When this linkage is disrupted through mutations in these glycotransferase genes multiple forms of congenital muscular dystrophy occur [408]. *O*-mannose-ribitol is a novel component of the phosphotrisaccharide linkage to xylose in α-DG (Figure 7). A CDP-ribitol ISPD (isoprenoid synthase domain) cytidyl transferase generates the reduced sugar nucleotide required for the insertion of ribitol in this phosphodiester linkage [406]. Sequential TMEM5 (UDP-xylosyl transferase), B4GAT1 (glucuronyl transferase), and LARGE (dual-function glycosyltransferase) enzymes modify this ribitol phosphodiester moiety to generate a functional α-DG receptor interactive with the LamG motifs of ECM proteins.

## 6. Concluding Remarks

This review has highlighted the complexities of HS and its differential processing in different biological tissues in health and disease. The use of well-defined synthetic 3-O-sulfated standards, comprehensive HS disaccharide profiling, and cell engineering has facilitated previously unappreciated differences in 3-O-sulfation profiles in clinical heparins and HS3ST-generated structural variation in cell surface HS. The availability of these standards has significantly improved the precision of the identification of these HS species. The sulfation of HS GAGs is an important functional determinant that imparts diverse regulatory properties to HS-PGs. The charge transfer properties conveyed by sulfate groups decorating HS GAGs represent the cell signalling and cell-ECM communication machinery that regulates cellular behaviour. Sulfated glycans and glycolipids regulate cell growth and differentiation, ECM synthesis and organization and mechanosensory signalling processes which modulate tissue homeostasis and repair and remodelling processes that occur in health and disease. Transduction of intrinsic biomechanical forces in the cellular microenvironment regulates cellular behaviour including stem cells in their niche environment through the involvement of cell-associated and pericellular HS-PGs. HS is a highly versatile GAG reflected in its wide tissue distribution. HS is packed with information that facilitates its diverse roles in molecular recognition and information transfer to cells, thus mediating cellular control over tissue development and wound repair as well as many other essential physiological processes. HS is known to be interactive with over 400 proteins listed in the HS interactome. However, a model of the actively inflamed murine pancreas has further extended the total number of known HS binding proteins to at least 786, which explains the wide diversity of ligand interactions and cellular behaviours that HS participates in. Information is steadily increasing on HS and its interactions in health and disease, and mimetic peptides based on some of these interactions are being evaluated in several potential therapies to control inflammation and promote tissue regeneration or as anti-tumour agents. HS-PGs continue to be identified and their functional properties identified, this class of PG will continue to evolve with the identification of further members. Well-executed systematic basic studies on HS and its diverse molecular and cellular interactions will bring clinical applications one step closer. Perlecan is found in the stem cell niche and promotes MSC differentiation into specific progenitor cell lineages. The identification of specific HS-PGs involved in driving hMSC lineage specification will likely provide new markers to allow a better selection of hMSCs and their expansion for use in therapeutic applications. Recent studies elucidating roles for HS-PGs in synaptic stabilization, neuronal photo-transductive network activity and assembly and function of the neuromuscular joint point to the diversity of neuroregulatory processes in which HS-PGs participate. A greater understanding of HS structure and function illustrates the potential of HS molecular interactions in the development of more effective therapeutic approaches to the treatment of specific disease processes. Sophisticated analytic procedures have been developed for the analysis of HS structure and function relationships not only in normal developmental tissues but also in diseased tissues and in the design of HS mimetics. This area of drug design has entered an exciting era with the advanced analytical techniques now available and will undoubtedly lead to significant improvement in HS therapeutics and their application in repair biology.

## Figures and Tables

**Figure 1 ijms-24-01148-f001:**
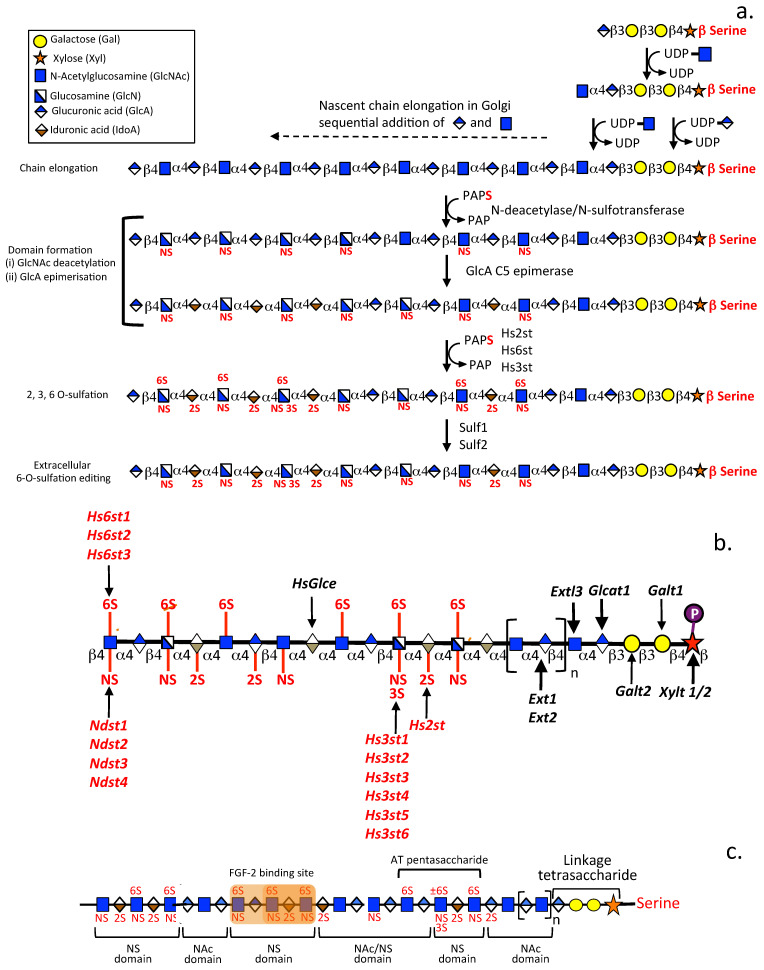
The multiple steps in the biosynthesis of putative heparin and HS chains (**a**), and the multiple enzymes that sulfate specific regions (**b**) and identification of specific HS sequences and sulfate presentations that provide FGF-2 and AT binding sites in a putative HS chain (**c**). Figure reprinted/adapted from Suflita M et al. [22] with the permission of Springer publishers. The GAG symbols used in this figure are those recommended by the symbol nomenclature for glycans (SNFG) discussion group [23].

**Figure 2 ijms-24-01148-f002:**
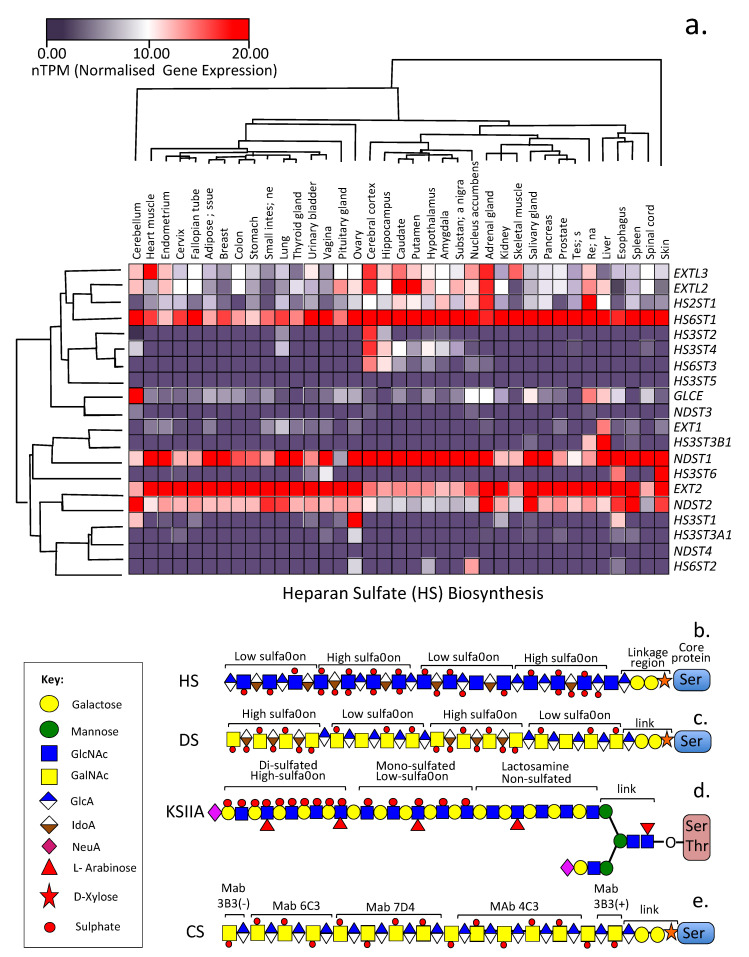
Variation in HS sulfation in a number of biological tissues and in GAG side chain structure. (**a**) Heatmap depicting the expression of twenty GAG biosynthetic enzymes demonstrating transcript expression levels for HS in 37 human tissues based on RNA sequencing (normalized transcripts per million, nTPM). Hierarchical clustering is shown based on the Pearson correlation. Data were obtained using the Genotype-Tissue Expression (GTEx) Program based on The Human Protein Atlas version 21.0 and Ensembl version 103.38. Figure modified from [24]. (**b**–**e**) Areas of high and low sulfation regions in putative GAG chains and specific regions in CS chains identified by a range of mAbs to CS sulfation motifs used in the assembly of CS chains. Figure 2a modified from Basu et al. [24] Reproduced under the Creative Commons Attribution CC-BY 4.0. with permission of The American Physiological Society.

**Figure 3 ijms-24-01148-f003:**
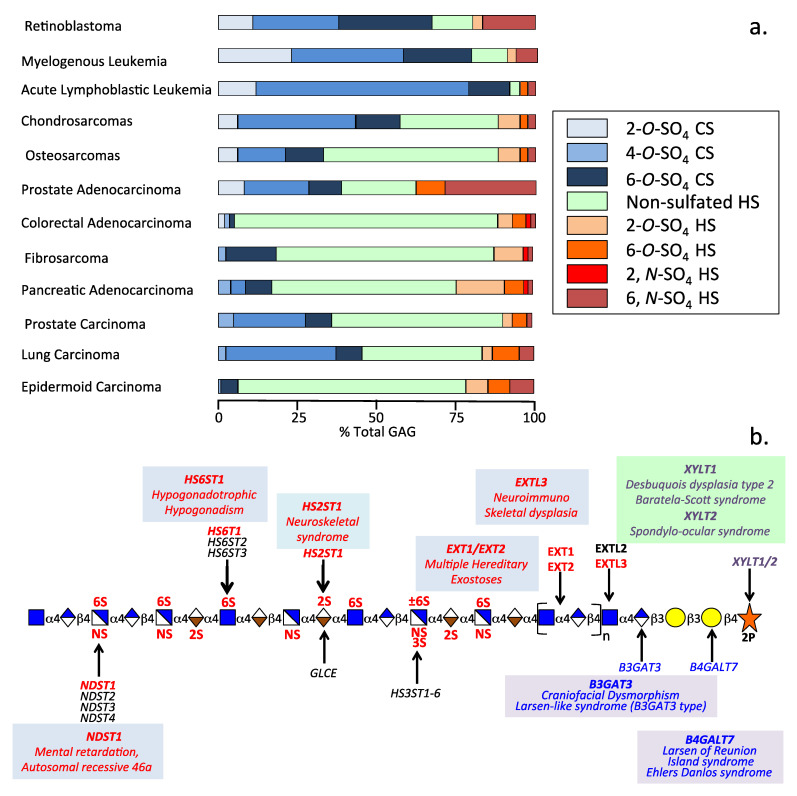
Variation in the sulfation patterns of CS and HS in cancer (**a**) and mutations in HS sulfation enzymes associated with several musculoskeletal disorders (**b**). Data modified from Basu et al. [24] Reproduced under the Creative Commons Attribution CC-BY 4.0. with permission of The American Physiological Society.

**Figure 5 ijms-24-01148-f005:**
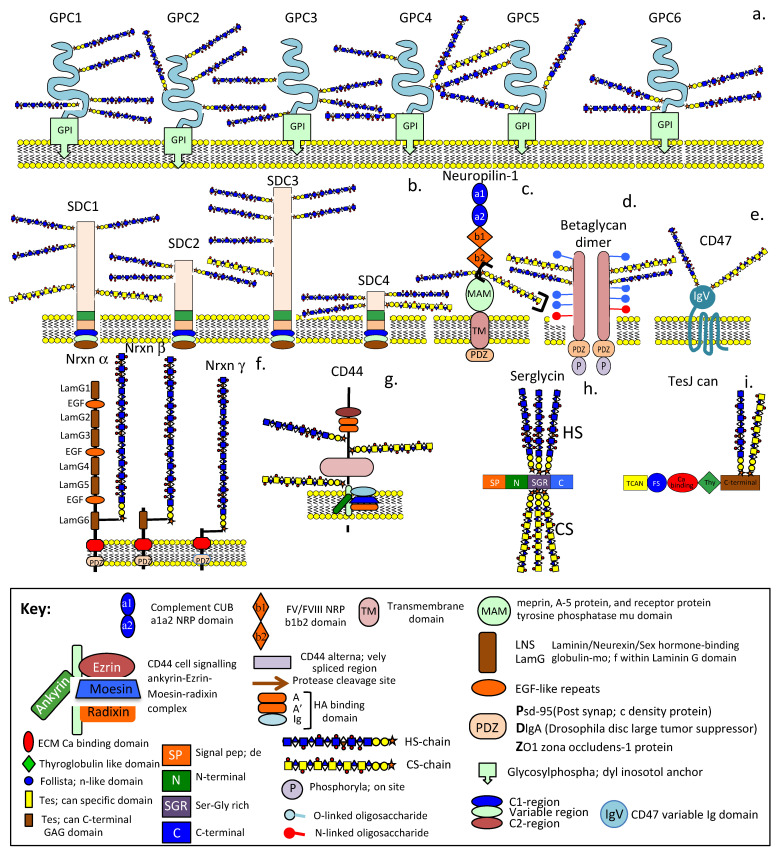
Cell surface and intracellular HS-proteoglycans. (**a**) the glycophospatidyl choline anchored cell surface glypican family, (**b**) the syndecan family, (**c**) cell surface neuropilin-1, (**d**) betaglycan, (**e**) CD47, (**f**) the synapse stabilising neurexins, (**g**) CD44, (**h**) serglycin, (**i**) testican. Neuropilin occurs as a proteoglycan containing a single HS or a single CS chain this is why the CS chain is bracketed to make this point. Testican (**i**) is a part time HS-PG and also contains CS chains. Mast cell serglycin is exclusively substituted with heparin however serglycin produced by other cell types can also contain CS chains.

**Figure 6 ijms-24-01148-f006:**
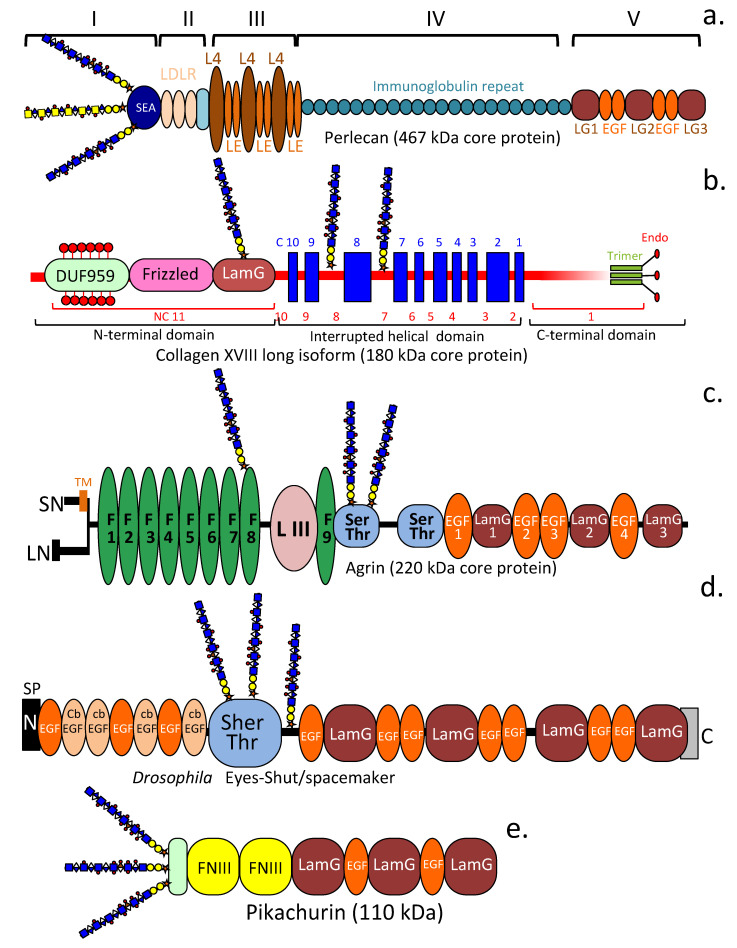
Extracellular HS-proteoglycans and their core protein domain organisations and GAG attachment sites. (**a**) perlecan, (**b**) agrin, (**c**) type XVIII collagen, (**d**) Eyes-shut (**e**) Pikachurin.

**Figure 7 ijms-24-01148-f007:**
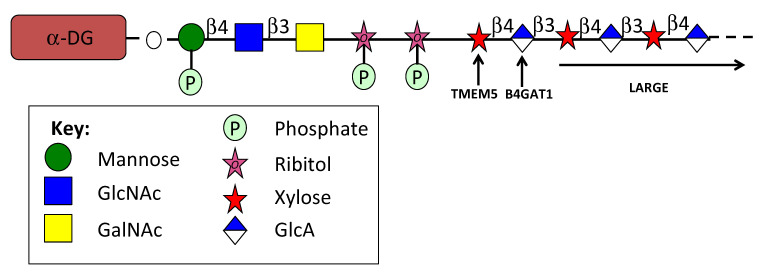
The α-DG LamG binding receptor and its constituent glycan components assembled from a GalNAc-β3-GlcNAc β4 Mannose M3 trisaccharide link module with sequential attachment of a number of glycan components by glycosyl transferases including TMEM5, B4GAT1, and the LARGE xylosyl and glucuronyl transferase complex to form a functional receptor. Phosphorylated ribitol is an unusual component in this assembly process. Figure re-drawn from data provided in [407,409].

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
