# Peer review of "HS, an Ancient Molecular Recognition and Information Storage Glycosaminoglycan, Equips HS-Proteoglycans with Diverse Matrix and Cell-Interactive Properties Operative in Tissue Development and Tissue Function in Health and Disease"

_ijms, 2023, doi:10.3390/ijms24021148_

Round 1

Reviewer 1 Report

This review is rich and rather well documented citing a lot of references but not always as targeted as could be. It would benefit from some expansion and reduction as suggested below.
The title is terrible and not evoking a review.
The third author is a ghost in the submitted version.

Specific comments:

lines 190-191 why cite all recently published human cell or tissue atlases?

lines 254-257: reference to bioinformatics is not properly addressed. Ref 84 is 1) a very incomplete coverage of the topic and 2) was updated recently: doi:10.1007/978-1-0716-1398-6_50 in a volume dedicated to GAGs. A much more comprehensive and recent review is a more instructive alternative: doi: 10.1016/j.tibs.2020.10.004 .
Furthermore, ref 85 is not the one to provide for illustrating the topic of a GAG interactome database but MatrixDB: doi: 10.1093/nar/gky1035, that is a particularly relevant resource in the context of GAG studies and is surprisingly absent despite its longevity (first release in 2009: doi: 10.1093/bioinformatics/btp025). This resource warrants several sentences in this review. Actually, the authors seem to have a broad definition of "database" as they cite articles actually not describing a database but datasets (refs 8 and 85). Nowadays, useful databases are on-line and specified with a URL. Line 295: ref 91 is by no means a database.

Line 332-354: Missed a recent remarkable technological development doi: 10.1126/sciadv.abl6026 and the mention of ion mobility based MS. Another promising technology for sequencing GAGs relies on lasers and infrared spectroscopy that is not mentioned. Both are   reviewed here doi: 10.1016/j.sbi.2018.06.006.
Other missed points:
- the CRISPR-Cas9 technology is slowly but surely being used to interfere with GAG biosynthesis thereby helping to understand it better
- GAG arrays are used in an increasing number of studies to chase precise sulfation patterns.

Lines 355-369: This section poorly reflects 3D modelling work produced for GAGs.
Another missed database: GAG-DB: doi: 10.3390/biom10121660 along with a missed interface:
GAG Builder: doi: 10.3390/biom10121660
More generally, 3D aspects of GAGs were reviewed in 2021 in doi: 10.3390/biom11050739 that should guide the authors on this topic.

Lines 370-382: This section is particularly brief and except for ref 130, cites rather random reviews that are not really relevant to the field of glycomics. The authors should rather consult these 2 recent reviews to find more appropriate arguments and citations:
doi: 10.1016/j.biotechadv.2022.108008
doi: 10.1021/acs.chemrev.2c00110

Lines 403-405: "Deciphering the complexity of ECM glycosylation signatures will be invaluable in the discovery of novel therapeutics and biomarkers in the diagnosis and treatment of diseases". This is more advanced than the references cited. See doi: 10.1073/pnas.2115328119 admittedly published after this manuscript was submitted but still a big step forward to be mentioned in the revised version.

Lines 409-421: This section is also limited. Definitely the Clausen group has published remarkable work on integrating transcriptomics data but they are not alone. See again the review mentioned above:  doi: 10.1016/j.tibs.2020.10.004

Lines 442-446: Naive readers may not know what is Gene Ontology KEGG. The bear minimum is to provide references to these two resources. The sentence does not really bring much information out of context. It seems just retranscribed from citations.

Section 4 and 5 as a whole are very enumerative. A lot of information contained in these section is textbook knowledge (e.g., Essentials of glycobiology, chapter 17) or are already covered in well regarded review journals such as Chem Rev doi: 10.1021/acs.chemrev.8b00354. So it would be more useful for the reader to be offered complement information based on already existing one. It is surprising for instance that the term "glycocalyx" is used in the abstract and in the first line of the introduction never to be seen again in the manuscript. The concept of glycocalyx is central and often unknown and yet not used as a unifying view in this review. It would be interesting to raise the attention of the reader on refs 108 and 244 that describe "perineuronal nets" that described a similar super-structure and hardly ever connected to descriptions of the glycocalyx when the role of GAGs seems comparable. As one of the authors has written ref 17 for example, it would seem natural to expand the discussion on the interesting role of PG fragments, for example. Enzymes are another poorly mentioned topic, etc.

There is no section 6.
The concluding remarks may have to be refined to reflect expected changes in the main parts of the manuscript.

Figures:
It would be relevant to mention that Figure 1 is depicted using the SNFG notation summarised in the legend. For newcomers to the field, this notation may appear cryptic and deserves a citation to SNFG doi: 10.1093/glycob/cwz045. For the sake of consistency, sulfations on each monosaccharide should be kept in red in Figure 3 as in Figure 1 and 4.
Figure 5 and 7 are not using the SNFG notation and the change is confusing the reader. There should be a more consistent (re)presentation of molecules all throughout and adopting the SNFG standard is highly recommended.

Author Response

New title for HS REVIEW

HS, an Ancient Molecular Recognition and Information Storage Glycosaminoglycan Motif, Equips HS-Proteoglycans with Diverse Matrix and Cell Interactive Properties Employed in Tissue Development and the Remodelling of Tissues in Health and Disease.

Reviewer comments

lines 190-191 why cite all recently published human cell or tissue atlases?

Author response

These items have been removed from the revised manuscript

Reviewer comments

lines 254-257: reference to bioinformatics is not properly addressed. Ref 84 is 1) a very incomplete coverage of the topic and 2) was updated recently: doi:10.1007/978-1-0716-1398-6_50 in a volume dedicated to GAGs. A much more comprehensive and recent review is a more instructive alternative: doi: 10.1016/j.tibs.2020.10.004 .
Furthermore, ref 85 is not the one to provide for illustrating the topic of a GAG interactome database but MatrixDB: doi: 10.1093/nar/gky1035, that is a particularly relevant resource in the context of GAG studies and is surprisingly absent despite its longevity (first release in 2009: doi: 10.1093/bioinformatics/btp025). This resource warrants several sentences in this review. Actually, the authors seem to have a broad definition of "database" as they cite articles actually not describing a database but datasets (refs 8 and 85). Nowadays, useful databases are on-line and specified with a URL. Line 295: ref 91 is by no means a database.
Author response

Your suggestions have been incorporated into the revised manuscript.

Reviewer comments

Line 332-354: Missed a recent remarkable technological development doi: 10.1126/sciadv.abl6026 and the mention of ion mobility based MS. Another promising technology for sequencing GAGs relies on lasers and infrared spectroscopy that is not mentioned. Both are   reviewed here doi: 10.1016/j.sbi.2018.06.006.
Other missed points:
- the CRISPR-Cas9 technology is slowly but surely being used to interfere with GAG biosynthesis thereby helping to understand it better
- GAG arrays are used in an increasing number of studies to chase precise sulfation patterns.
Author responses

Your suggestions have been incorporated into the revised manuscript.  We have also added additional comments on CRISPR-Cas9 and transcriptomics to the revision

Reviewer comments
Lines 355-369: This section poorly reflects 3D modelling work produced for GAGs.
Another missed database: GAG-DB: doi: 10.3390/biom10121660 along with a missed interface:
GAG Builder: doi: 10.3390/biom10121660
More generally, 3D aspects of GAGs were reviewed in 2021 in doi: 10.3390/biom11050739 that should guide the authors on this topic.

Author responses

Your suggestions have been incorporated into the revised manuscript. 

Reviewer comments

Lines 370-382: This section is particularly brief and except for ref 130, cites rather random reviews that are not really relevant to the field of glycomics. The authors should rather consult these 2 recent reviews to find more appropriate arguments and citations:
doi: 10.1016/j.biotechadv.2022.108008
doi: 10.1021/acs.chemrev.2c00110

Author responses

Your suggestions have been incorporated into the revised manuscript. 

Reviewer comments
Section 4 and 5 as a whole are very enumerative. A lot of information contained in these section is textbook knowledge (e.g., Essentials of glycobiology, chapter 17) or are already covered in well regarded review journals such as Chem Rev doi: 10.1021/acs.chemrev.8b00354. So it would be more useful for the reader to be offered complement information based on already existing one. It is surprising for instance that the term "glycocalyx" is used in the abstract and in the first line of the introduction never to be seen again in the manuscript. The concept of glycocalyx is central and often unknown and yet not used as a unifying view in this review. It would be interesting to raise the attention of the reader on refs 108 and 244 that describe "perineuronal nets" that described a similar super-structure and hardly ever connected to descriptions of the glycocalyx when the role of GAGs seems comparable. As one of the authors has written ref 17 for example, it would seem natural to expand the discussion on the interesting role of PG fragments, for example. Enzymes are another poor

Author response

 We have removed information on the established HS-PGs and provided a reference list for further reading concentrating only on the newer HS-PGs.  We have elaborated on your comments on the glycocalyx with an additional segment added to the revised manuscript.

Our original intention was to include a section on PG fragments and enzymes but at that time the manuscript was getting too big to include all this information and we made a decision to reserve this information for a follow up review when time permits.

Reviewer comments
Lines 403-405: "Deciphering the complexity of ECM glycosylation signatures will be invaluable in the discovery of novel therapeutics and biomarkers in the diagnosis and treatment of diseases". This is more advanced than the references cited. See doi: 10.1073/pnas.2115328119 admittedly published after this manuscript was submitted but still a big step forward to be mentioned in the revised version.

Author responses

Your suggestions have been incorporated into the revised manuscript. 

Reviewer comments

Lines 409-421: This section is also limited. Definitely the Clausen group has published remarkable work on integrating transcriptomics data but they are not alone. See again the review mentioned above:  doi: 10.1016/j.tibs.2020.10.004
Author responses

Additional comments have been added to the revised manuscript.

Reviewer comments

Lines 442-446: Naive readers may not know what is Gene Ontology KEGG. The bear minimum is to provide references to these two resources. The sentence does not really bring much information out of context. It seems just retranscribed from citations.

Author response

Information has been added on KEGG in the revised manuscript

Reviewer comments

There is no section 6.
The concluding remarks may have to be refined to reflect expected changes in the main parts of the manuscript.

Author response

Modified in the revised manuscript

Reviewer comments
Figures:
It would be relevant to mention that Figure 1 is depicted using the SNFG notation summarised in the legend. For newcomers to the field, this notation may appear cryptic and deserves a citation to SNFG doi: 10.1093/glycob/cwz045. For the sake of consistency, sulfations on each monosaccharide should be kept in red in Figure 3 as in Figure 1 and 4.
Figure 5 and 7 are not using the SNFG notation and the change is confusing the reader. There should be a more consistent (re)presentation of molecules all throughout and adopting the SNFG standard is highly recommended.

Author response

All figures have been amended in the revised manuscript taking your recommendations on board and comments made by reviewer 2.

Reviewer comment

It is surprising for instance that the term "glycocalyx" is used in the abstract and in the first line of the introduction never to be seen again in the manuscript. The concept of glycocalyx is central and often unknown and yet not used as a unifying view in this review. It would be interesting to raise the attention of the reader on refs 108 and 244 that describe "perineuronal nets" that described a similar super-structure and hardly ever connected to descriptions of the glycocalyx when the role of GAGs seems comparable. As one of the authors has written ref 17 for example, it would seem natural to expand the discussion on the interesting role of PG fragments, for example. Enzymes are another poorly mentioned topic, etc.
Author response

In the revised manuscript we have added a segment further exploring the concept of the glycocalyx and how HS-PGs contribute to synaptic neuronal cellular responses in what we are proposing as a sophisticated specialized glycocalyx with specific roles in normal tissue development relevant to cellular responses in tissue homeostasis and in neurodegenerative pathology.  Co-ordinated cell mediated responses provided by HS-and CS-PGs as a molecular switch is also explored as a regulatory mechanism.

Reviewer comment

Figures:
It would be relevant to mention that Figure 1 is depicted using the SNFG notation summarised in the legend. For newcomers to the field, this notation may appear cryptic and deserves a citation to SNFG doi: 10.1093/glycob/cwz045. For the sake of consistency, sulfations on each monosaccharide should be kept in red in Figure 3 as in Figure 1 and 4.
Figure 5 and 7 are not using the SNFG notation and the change is confusing the reader. There should be a more consistent (re)presentation of molecules all throughout and adopting the SNFG standard is highly recommended.

Author response

We have re-done the figures employing the SNFG iconology and provided additional information as suggested and cited the SNFG reference.  We originally did not use this symbology since we were concerned that this information could not be adequately portrayed due to size constraints on the figures.  However we feel the information provided in the new figures is portrayed quite nicely and thank you for your suggestion.

Reviewer 2 Report

This is a timely review of proteoglycans, with emphasis on species carrying heparan sulfate (HS) side chains. The approach is comprehensive in covering structural, functional, pathophysiological, as well as methodological aspects, and updates information regarding several recently discovered proteoglycan species. The following issues should be considered to further improve the quality of the report.

Major issues

1.     The functional importance of sulfation patterns forms a major theme throughout the report, repeatedly referred to as the ‘sulfation code’. This is a misleading expression, in the sense that it conveys the impression of HS products with strictly distinct sulfation patterns. The enzymatic polymer-modification reactions involved are stochastic, leading to heterogeneous mixtures of HS chains, yet with average compositions regulated in cell-autonomous fashion. These features allow for graded modulation of protein functional response. As correctly noted by the authors (lines 182-185), HS polymer modification appears regulated at the transcriptional level such that the enzymes involved are presented at varying relative amounts. Recent findings suggest that their reactions are controlled through concentration-dependent enzyme complex formation. In my opinion, the presentation would gain by introducing these concepts. References for consideration: Lindahl and Li, Int Rev Cell Mol Biol. 2009, 276:105-59; Kjellén and Lindahl, Curr Opin Struct Biol. 2018, 50:101-108; Zhang et al., Carbohydrate Polymers 2023, 299 - https://doi.org/10.1016/j.carbpol.2022.120191.

2.     Several aspects of the figures require further attention. Fig. 1a - The scheme indicated points toward biosynthesis of heparin rather than HS. The structural difference between heparin and HS should be explained in the text (see also point 3). Fig. 1c – The dashed frame around the ‘FGF-2 binding site’ is confusing; the same frame occurs in Fig. 4d, where it is said to represent the site for combined binding of FGF2 and FGFR. The orange-colored trisaccharide, according to Fig. 4 exposes binding sites for FGFR as well as FGF2; however, the minimal binding site for FGF2 according to literature (Guglier et al., Biochemistry 2008, and others) is either a penta- or a tetrasaccharide. Please clarify. Fig. 1d is mentioned on line 145 – but no panel (d) seen in the actual figure. Fig. 5b – the testican structure should be shown to contain CS chains? Fig. 5 – pikachurin (panel l) is missing in figure but mentioned in the figure legend. Fig. 6 – the color codes need to be better explained; as presented, this figure makes little sense. Fig. 7 – description of (e) and (f) missing in subtitle.

Minor issues

3.     Lines 132-134 – This statement is an oversimplification. There are HS species that are less sulfated than CS/DS. Further, N-acetylated GlcNAc residues in heparin occur as single entities, not in ‘high acetylation’ domains.

4.     Lines 223 - 226 – This sentence gives the impression that the Sulf enzymes can modify CS chains; the actual target is HS.

5.     Line 295-298 – This general statement does not reflect the topics of refs 92 and 93, that are restricted to regulation of Hs3st.

6.     Line 455-457 – Is antithrombin an ‘ECM protein’?

7.     Lines 505-506 – This sentence should be rephrased.

8.     Lines 610-623 - Neuropilin is listed as one of the cell-surface HS-PGs, as shown also in Fig. 4. The HS/CS chains should be mentioned in the text.

9.     Lines 616 – 618 – The text refers to ‘heparin’; should it rather be ‘heparin oligosaccharides’?

10.  Are all acronyms needed?

11.  The Abstract should be condensed.

Author Response

Point by Point Responses to Reviewer 2 appraissal

Comments and Suggestions for Authors

This is a timely review of proteoglycans, with emphasis on species carrying heparan sulfate (HS) side chains. The approach is comprehensive in covering structural, functional, pathophysiological, as well as methodological aspects, and updates information regarding several recently discovered proteoglycan species. The following issues should be considered to further improve the quality of the report.

Major issues

Reviewer comment

The functional importance of sulfation patterns forms a major theme throughout the report, repeatedly referred to as the ‘sulfation code’. This is a misleading expression, in the sense that it conveys the impression of HS products with strictly distinct sulfation patterns. The enzymatic polymer-modification reactions involved are stochastic, leading to heterogeneous mixtures of HS chains, yet with average compositions regulated in cell-autonomous fashion. These features allow for graded modulation of protein functional response. As correctly noted by the authors (lines 182-185), HS polymer modification appears regulated at the transcriptional level such that the enzymes involved are presented at varying relative amounts. Recent findings suggest that their reactions are controlled through concentration-dependent enzyme complex formation. In my opinion, the presentation would gain by introducing these concepts. References for consideration: Lindahl and Li, Int Rev Cell Mol Biol. 2009, 276:105-59; Kjellén and Lindahl, Curr Opin Struct Biol. 2018, 50:101-108; Zhang et al., Carbohydrate Polymers 2023, 299 - https://doi.org/10.1016/j.carbpol.2022.120191.

Author response

The following segment has been added to the revised manuscript to address you point.

While distinct HS functional motifs have been identified, HS biosynthetic reactions have an innate ability to produce variable HS structural forms which share interactive properties through ionic interactions involving their negatively charged sulfate and carboxylate groups. Thus while some protein ligand interactions may involve strictly defined HS structures other interactions also occur with no apparent requirement for distinct saccharide sequence for such interactions. This raises questions of how essential the control of production of these specific sulfation motifs are when more widespread HS interactions also occur [1]. Binding affinity and specificity are determined by charge distribution of the sulfate and carboxylate groups and how these are presented in planar orientations [2]. Sulfate groups are bulky entities and the correct 3D spatial presentation in some cases may be an important requirement for effective receptor interactions. Epimerization of GlcA to IdoA introduces greater flexibility in the HS backbone which may allow a more extensive exploration of sulfate groups in various orientations with interactive partners. Some HS interactions however may be nonspecific and ionic in nature. Thus charge density in tissues and highly specific interactions involving rare HS-structures should both be considered in the control of tissue development, HS biosynthesis in tissues is a stochastic process.

  1. Lindahl U, Li JP. Interactions between heparan sulfate and proteins-design and functional implications. Int Rev Cell Mol Biol. 2009;276:105-59.
  2. Kjellén L, Lindahl U. Specificity of glycosaminoglycan-protein interactions. Curr Opin Struct Biol. 2018 Jun;50:101-108.

Reviewer comment

Several aspects of the figures require further attention. Fig. 1a - The scheme indicated points toward biosynthesis of heparin rather than HS. The structural difference between heparin and HS should be explained in the text (see also point 3). Fig. 1c – The dashed frame around the ‘FGF-2 binding site’ is confusing; the same frame occurs in Fig. 4d, where it is said to represent the site for combined binding of FGF2 and FGFR. The orange-colored trisaccharide, according to Fig. 4 exposes binding sites for FGFR as well as FGF2; however, the minimal binding site for FGF2 according to literature (Guglier et al., Biochemistry 2008, and others) is either a penta- or a tetrasaccharide. Please clarify. Fig. 1d is mentioned on line 145 – but no panel (d) seen in the actual figure. Fig. 5b – the testican structure should be shown to contain CS chains? Fig. 5 – pikachurin (panel l) is missing in figure but mentioned in the figure legend. Fig. 6 – the color codes need to be better explained; as presented, this figure makes little sense. Fig. 7 – description of (e) and (f) missing in subtitle.

Author response

The figures have been modified to satisfy your requests and additional explanatory comments added to the legends. Fig 4 has been modified to better explain what is known about FGF-2/FGFR binding. Figure 6 has been removed from the revised manuscript.

Minor issues

Reviewer comment

Lines 132-134 – This statement is an oversimplification. There are HS species that are less sulfated than CS/DS. Further, N-acetylated GlcNAc residues in heparin occur as single entities, not in ‘high acetylation’ domains.

Author response

Comments have been added to the revised manuscript to address these points

Reviewer comment

Lines 223 - 226 – This sentence gives the impression that the Sulf enzymes can modify CS chains; the actual target is HS.

Author response

This has been rectified in the revised manuscript

Reviewer comment

Line 295-298 – This general statement does not reflect the topics of refs 92 and 93, that are restricted to regulation of Hs3st.

Author response

This has been modified in the revised manuscript

Reviewer comment

Line 455-457 – Is antithrombin an ‘ECM protein’?

Author response

Antithrombin is a cell surface protein

Reviewer comment

Lines 505-506 – This sentence should be rephrased.

Author response

MDPI did not supply me a manuscript with line numbers so I could not ascertain the sentence you are referring to.

Reviewer comment

Lines 610-623 - Neuropilin is listed as one of the cell-surface HS-PGs, as shown also in Fig. 4. The HS/CS chains should be mentioned in the text.

Author response

A segment has been added to the revised manuscript.

A substantial proportion of NRP1 is a proteoglycan modified with either HS or CS on a single conserved Ser residue in the core protein. GAG free forms of NRP-1 are also common. The composition of the NRP1 GAG chains differ between endothelial cells and smooth muscle cells however a single GAG chain cannot contain both HS and CS simultaneously, thus endogenous NRP1 exists as either an HS-PG or CS-PG but not as a hybrid proteoglycan. The Ser residue that is substituted with GAG is located between the b1b2 and MAM domains of NRP1 and is conserved between vertebrates. NRP2, a mammalian homolog of NRP1, does not have this conserved Ser residue [252].

Reviewer comment Lines 616 – 618 – The text refers to ‘heparin’; should it rather be ‘heparin oligosaccharides’?

Author response

Comment modified in the revised manuscript

Reviewer comment Are all acronyms needed?

Author response

These have been significantly reduced in the revised manuscript

Reviewer comment

The Abstract should be condensed.

Round 2

Reviewer 1 Report

The review has significantly improved but a few issues remain. Mainly, a proportion of references remains loosely cited and do not add to the discussion about GAGs. Almost the opposite, it's adding noise to the set of references. So unless these references are dug into and the parts relevant to studying GAGs and PGs are commented in the manuscript, there is no point in citing them. There are examples below.

The harmonisation of figures is a real plus.

Prior to listing details, it is really surprising not to see the reference manual Essentials of Glycobiology (https://www.ncbi.nlm.nih.gov/books/NBK579918/) cited for basic definitions used in the introduction. This was mentioned in the previous set of comments and at least Chapter 17 should be referred to.

line 231: I maintain that ref 17, 90 and 91 describe dataSETs not dataBASEs. These references provide tables with data but these are by no means databases ( see: https://en.wikipedia.org/wiki/Database)

line 257: what is glycan substitution? ref 116 is dedicated to describing glycan ligands. Maybe the reference manual could be cited again here?

line 261: the Clausen of ref 118 is not Henrik... this citation is out of scope and not related to glycosylation

line 266: lumping together ref 123 to 131 does not make sense and does not provide the reader with accurate information. Please read ref 127 to convince yourselves of the wealth of information available in glycoinformatics. They are multiple angles and levels and this is not properly conveyed here. In any event, it is not the purpose of this review to describe these aspect and the authors should stick, as much as possible, to referring to GAG and PG related resources which are extremely limited as a matter of fact.That's why ref 127 is convenient to hide behind.

lines 267-291: my comment on KEGG was wrongly understood and the importance given to this resource in the revised version is disproportionate compared to for instance MatrixDB (now refs 124, 125) which is much more topical and apparently does not deserve to be called by its name and outlined in the text.The only needed information was a reference to KEGG now well represented in ref 134-137. The part on genes and genomes (lines 280-287) brings nothing to this review. And, there seems to be a misunderstood contribution of Gene Ontology that is a 100% KEGG-independent initiative (see geneontology.org). KEGG uses GO; that's all. So a reference to what Gene Ontology is, is needed.

line 326-27: the key discussion regarding the use of standards that is just mentioned in relation to ref 144 should be extended in the conclusion as it is a major hurdle that hampers progress in systematising the GAG field.

lines 365-66: AI-related papers on drug discovery are cited and they are only reviews in which the appraisal of drug discovery hardly ever worries about GAGs. Ref 126 has 1 remote ref to carbohydrates in a collection 482 but it is in fact a very general overview that is irrelevant to the present concern about GAGs. Same with ref 129, 130 131, which makes the sentence misleading.

line 367-68: the sentence seems to say that ref 157,158 are dedicated to carbohydrate synthesis when both are treating all chemical molecules at the same level.

line 463-64: same situation again, ref 216, 218,219,220 that have nothing to do with GAGs are lumped together with others that do mention GAGs or PGs as part of a misleading sentence.

This is in stark contrast with very adequately cited references in revised sections 3.3-5.

Minor:
- please check line 714 for possible truncation
- the number of sections is again not clear. There is a ghost Section 5 in between Section 4 and 6 but not called as such. The precise numbering of Section 2, 3 and 4 is dropped for 5 but should re-established.

Author Response

Reviewer 1 comments round 2.

Reviewer comment

The review has significantly improved but a few issues remain. Mainly, a proportion of references remains loosely cited and do not add to the discussion about GAGs. Almost the opposite, it's adding noise to the set of references. So unless these references are dug into and the parts relevant to studying GAGs and PGs are commented in the manuscript, there is no point in citing them. There are examples below.
Author response

We have deleted all references you have pointed out in the revised manuscript.

Reviewer comment

Prior to listing details, it is really surprising not to see the reference manual Essentials of Glycobiology (https://www.ncbi.nlm.nih.gov/books/NBK579918/) cited for basic definitions used in the introduction. This was mentioned in the previous set of comments and at least Chapter 17 should be referred to.
Author response

The Essentials of Glycobiology ref is now cited in the introduction.

Reviewer comment
line 231: I maintain that ref 17, 90 and 91 describe dataSETs not dataBASEs. These references provide tables with data but these are by no means databases ( see: https://en.wikipedia.org/wiki/Database)
Author response

I have deleted ref 17, 90, 91

Reviewer comment
line 257: what is glycan substitution? ref 116 is dedicated to describing glycan ligands. Maybe the reference manual could be cited again here?
Author response

Ref 116 is removed and the reference manual Essentials of Glycobiology (https://www.ncbi.nlm.nih.gov/books/NBK579918/) is cited here.

Reviewer comment
line 261: the Clausen of ref 118 is not Henrik... this citation is out of scope and not related to glycosylation

Author response
We have removed ref 118 from the revised manuscript.

Reviewer comment

line 266: lumping together ref 123 to 131 does not make sense and does not provide the reader with accurate information. Please read ref 127 to convince yourselves of the wealth of information available in glycoinformatics. They are multiple angles and levels and this is not properly conveyed here. In any event, it is not the purpose of this review to describe these aspect and the authors should stick, as much as possible, to referring to GAG and PG related resources which are extremely limited as a matter of fact. That's why ref 127 is convenient to hide behind.
Author response

we have deleted all refs except ref 127

Reviewer comment
lines 267-291: my comment on KEGG was wrongly understood and the importance given to this resource in the revised version is disproportionate compared to for instance MatrixDB (now refs 124, 125) which is much more topical and apparently does not deserve to be called by its name and outlined in the text. The only needed information was a reference to KEGG now well represented in ref 134-137. The part on genes and genomes (lines 280-287) brings nothing to this review. And, there seems to be a misunderstood contribution of Gene Ontology that is a 100% KEGG-independent initiative (see geneontology.org). KEGG uses GO; that's all. So a reference to what Gene Ontology is, is needed.
Author response

We have reduced our comments on KEGG as requested and added a few comments on the gene ontology consortium.

Reviewer comment

line 326-27: the key discussion regarding the use of standards that is just mentioned in relation to ref 144 should be extended in the conclusion as it is a major hurdle that hampers progress in systematising the GAG field.

Author response

We have added the comment This review has highlighted the complexities of HS and its differential processing in different biological tissues in health and disease. The use of well defined synthetic 3-O-sulfated standards, comprehensive HS disaccharide profiling, and cell engineering has facilitated previously unappreciated differences in 3-O-sulfation profiles in clinical heparins and HS3ST generated structural variation in cell surface HS. The availability of these standards has significantly improved the precision of identification of these HS species. To the Conclusions section.

Reviewer comment
lines 365-66: AI-related papers on drug discovery are cited and they are only reviews in which the appraisal of drug discovery hardly ever worries about GAGs. Ref 126 has 1 remote ref to carbohydrates in a collection 482 but it is in fact a very general overview that is irrelevant to the present concern about GAGs. Same with ref 129, 130 131, which makes the sentence misleading.
Author response

These less relevant refs you have pointed out have been removed from the manuscript

Reviewer comment
line 367-68: the sentence seems to say that ref 157,158 are dedicated to carbohydrate synthesis when both are treating all chemical molecules at the same level.
Author response

Refs this comment has been amended

Reviewer comment
line 463-64: same situation again, ref 216, 218,219,220 that have nothing to do with GAGs are lumped together with others that do mention GAGs or PGs as part of a misleading sentence.
Author response

Refs 216, 218, 219 , 220 are now deleted

Reviewer comment
This is in stark contrast with very adequately cited references in revised sections 3.3-5.

Minor:
- please check line 714 for possible truncation
- the number of sections is again not clear. There is a ghost Section 5 in between Section 4 and 6 but not called as such. The precise numbering of Section 2, 3 and 4 is dropped for 5 but should re-established.
Author response

Section numbering has been corrected.